# A shared transcriptional code orchestrates temporal patterning of the central nervous system

Andreas Sagner[1,2,3]*, Isabel Zhang[1], Thomas Watson[1], Jorge Lazaro[1¤], Manuela Melchionda[1], James Briscoe[1]*

1 The Francis Crick Institute, London, United Kingdom, 2 Faculty of Biology, Medicine and Health, The University of Manchester, Manchester, United Kingdom, 3 Institut für Biochemie, Emil-Fischer-Zentrum, Friedrich-Alexander-Universität Erlangen-Nürnberg, Erlangen, Germany

¤ Current address: European Molecular Biology Laboratory (EMBL) Barcelona, Barcelona, Spain
* andreas.sagner@fau.de (AS); james.briscoe@crick.ac.uk (JB)

**Data Availability Statement:** Sequencing data has been obtained from sources indicated in the Experimental Procedures section provided in the manuscript. Custom computer code is provided as

## Abstract

The molecular mechanisms that produce the full array of neuronal subtypes in the vertebrate nervous system are incompletely understood. Here, we provide evidence of a global temporal patterning program comprising sets of transcription factors that stratifies neurons based on the developmental time at which they are generated. This transcriptional code acts throughout the central nervous system, in parallel to spatial patterning, thereby increasing the diversity of neurons generated along the neuraxis. We further demonstrate that this temporal program operates in stem cell−derived neurons and is under the control of the TGFβ signaling pathway. Targeted perturbation of components of the temporal program, Nfia and Nfib, reveals their functional requirement for the generation of late-born neuronal subtypes. Together, our results provide evidence for the existence of a previously unappreciated global temporal transcriptional program of neuronal subtype identity and suggest that the integration of spatial and temporal patterning mechanisms diversifies and organizes neuronal subtypes in the vertebrate nervous system.

## Introduction

In mammals, the function of the nervous system depends on hundreds of molecularly and functionally distinct cell types [1]. This diversity requires the generation of different neuronal subtypes at the right place, time, and quantity during development. In turn, this guides the wiring of functioning neural circuits. The molecular mechanisms that direct the specification of distinct neuronal classes at characteristic positions, by subdividing the developing nervous system into topographical territories, have received considerable attention [2,3]. However, spatial patterning programs are not sufficient to account for the diversity of neuronal subtypes observed in the nervous system. Even within the same region of the nervous system, most neuronal classes can be further partitioned into distinct subtypes based on molecular and functional properties [4–9].

Supplemental Experimental Procedures and available via the Github repository as stated in the manuscript. All other relevant data are within the paper and its Supporting Information files.

**Funding:** This work was funded by the Francis Crick Institute (to JB), which receives its core funding from Cancer Research UK, the UK Medical Research Council and Wellcome Trust (all under FC001051); and by the European Research Council under European Union (EU) Horizon 2020 research and innovation program grant 742138 (to JB). AS was supported by the Human Frontier Science Program (LTF000401/2014-L), the University of Manchester Presidential Fellowship and by the Deutsche Forschungsgemeinschaft (DFG, German Research Foundation) – Projektnummer 455354162. Cancer Research UK (CRUK) supported IZ (C157/A23459). The funders had no role in study design, data collection and analysis, decision to publish, or preparation of the manuscript.

**Competing interests:** The authors have declared that no competing interests exist.

**Abbreviations:** ES, embryonic stem; Gdf11, Growth differentiation factor 11; H3K27ac, Histone-3-Lysine-27-acetylation; LMCm, medial lateral motor column; MN, motor neuron; RA, retinoic acid; RT-qPCR, real-time quantitative polymerase chain reaction; SAG, Shh pathway agonist; scRNAseq, single-cell RNA sequencing; Shh, Sonic Hedgehog; TF, transcription factor; TH, tyrosine hydroxylase; UMAP, Uniform Manifold Approximation and Projection.

Temporal mechanisms—the sequential production of different cell types at the same location—have been proposed to contribute to the generation of cell type diversity [10,11]. In the *Drosophila* nervous system, individual neuroblasts produce a characteristic temporal series of distinct neuronal subtypes [12–15]. Similar mechanisms have been documented in various regions of the vertebrate nervous system [11,16,17]. In the cortex, distinct subtypes of glutamatergic neurons are sequentially generated [18–20], in the hindbrain, first motor neurons (MNs) and later serotonergic neurons are generated from the same set of progenitors [21], while in the midbrain, the production of ocular MNs is followed by red nucleus neurons [22]. Moreover, progenitors throughout the nervous system typically produce neurons first and later generate glial cells such as astrocytes and oligodendrocytes [23,24]. However, whether temporal programs are a universal feature of neuronal subtype specification in the vertebrate nervous system and whether these are implemented by common or location specific gene expression programs is unclear.

The vertebrate spinal cord is an experimentally tractable system to address the basis of neuronal diversity. In this region of the nervous system, neurons process sensory inputs from the periphery relaying the information to the brain or to motor circuits that control and coordinate muscle activity. The temporally stratified generation of some of these neuronal subtypes has been documented, including inhibitory and excitatory neurons located in the dorsal horn as well as ventral motor and interneurons [25–32]. Nevertheless, a comprehensive picture is lacking, and the genetic programs that orchestrate the temporal patterning of the spinal cord are largely unclear. To this end, we recently characterized the emergence of neuronal diversity in the embryonic spinal cord [33]. This revealed sets of transcription factors (TFs), expressed at characteristic time points during the neurogenic period of spinal cord development, which further partition all major neuronal subtypes (Fig 1A). In all domains, the earliest neurons express Onecut family TFs, intermediate neurons express Pou2f2 and Zfhx2-4, while at late stages, subsets of neurons start to express Nfia/b/x, Neurod2/6, and Tcf4 [33–35]. This suggested the existence of a previously unappreciated temporal dimension to neuronal subtype generation in the spinal cord.

Although the role of these TFs had not been conceptualized as part of a globally coordinated temporal code, some have been implicated in the specification of neuronal subtypes. Onecut TFs, for example, are required in early-born V1 and MNs for the specification of Renshaw cells and medial lateral motor column (LMCm) neurons, respectively [31,36]. Onecut TFs and Pou2f2 also control the distribution of neurons from multiple dorsal–ventral domains [37–39]. Neurod2/6 control neuropeptide expression in inhibitory neurons in the dorsal horns of the spinal cord [40], and characterization of V2a neuron heterogeneity revealed that Zfhx3 and Neurod2/Nfib divide this neuronal class into a lateral and medial population [29]. Recent evidence further suggests that Zfhx3 and Nfib/Neurod2 partition neurons in the spinal cord into long-range projection and local interneurons [41]. Similar to the spinal cord, Onecut, Pou2f2, and Nfi-TFs label early and late-born neuronal subtypes in the retina and are required for their generation [42–44]. Zfhx3 and Nfi TFs also define distinct subpopulations of neurons generated by the midbrain floor plate, including dopaminergic neurons [45], and in the cerebral cortex [46]. Furthermore, Nfi TFs have also previously been identified as core components of a neurogenic transcriptional network in adult neural stem cells [47]. These observations raise the possibility that this temporal TF code is conserved in large parts of the central nervous system.

TGFβ signaling has been implicated in the timing of developmental temporal switches in the nervous system [48,49]. The transition from MN to serotonergic neurons and from ocular MNs to red nucleus neurons is accelerated by TGFβ signaling [48]. TGFβ signaling also promotes the expression of the late progenitor marker Nfia in neurogenic neural stem cells [50].

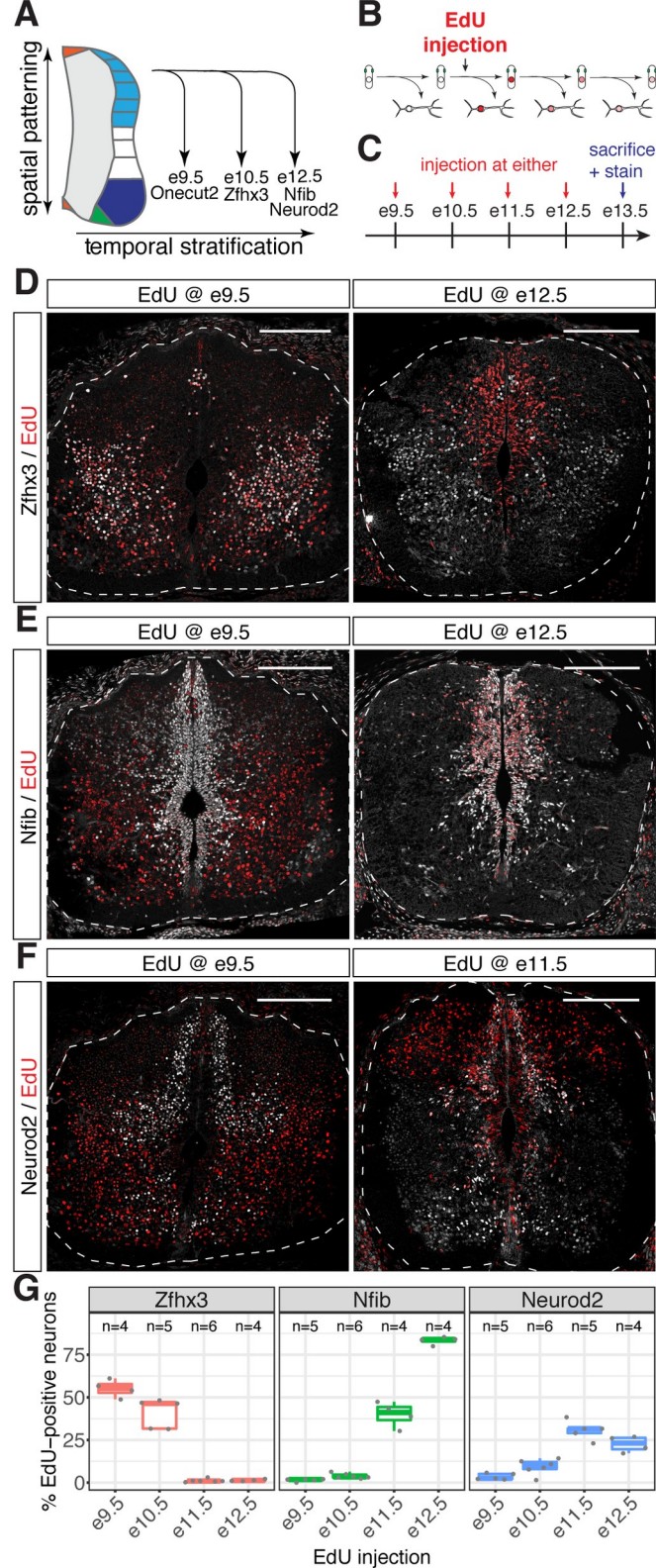

**Fig 1. Distinct birthdates of neurons expressing different temporal TFs (see also S1 and S2 Figs).** (A) Distinct cohorts of TFs are induced at different developmental stages in neurons from all dorsal–ventral domains in the spinal cord. (B) Scheme depicting EdU birthdating of neurons. (C) Dams were injected with EdU at e9.5, e10.5, e11.5, or

e12.5 and embryos collected at e13.5. Colocalization between EdU and temporal TFs was then assessed in spinal cord cryosections. (D) Zfhx3-positive neurons are labeled by EdU, when EdU is administered at e9.5, but not at e12.5. (E) EdU labels Nfib-positive neurons when administered at e12.5, but not at e9.5. (F) Neurod2-postive neurons are labeled when EdU is administered at e11.5, but not when EdU is administered at e9.5. (G) Percentage of EdU-positive neurons labeled by Zfhx3, Nfib, and Neurod2 in the spinal cord. Underlying data are provided in S1 Data. Scale bars in D, E, and F = 200 μm. TF, transcription factor.

Furthermore, Growth differentiation factor 11 (Gdf11), a ligand of the TGFβ family that signals via Activin receptors [51,52], has been implicated in the timing of MN subtype generation and onset of gliogenesis in the spinal cord [53]. TGFβ signaling is also important for controlling the timing of fate switches in the *Drosophila* nervous system [49], raising the possibility that it may serve as a general timer for the sequential generation of cellular subtypes.

Here, we demonstrate by EdU birthdating that a set of TFs comprise a temporal TF code that identifies neurons based on their time point of birth. We find that the same sequence of TF expression applies throughout the brain, including the forebrain, midbrain, hindbrain, and retina, and for stem cell–derived in vitro generated neurons with defined dorsal–ventral and axial identities. We also document a temporal patterning code for progenitors throughout the nervous system and provide evidence that TGFβ signaling controls the pace of the temporal program. Finally, to characterize the genetic programs that control the temporal specification of neurons, we perturb the function of the TFs Nfia and Nfib and show that their activity is required for the generation of late neuronal subtypes. Taken together, our data reveal conserved temporal patterning of neurons and progenitors in large parts of the nervous system that is under the control of the TGFβ signaling pathway and suggest a close link between the developmental programs that control the switch from neuro- to gliogenesis and the specification of neuronal diversity.

## Results

### EdU birthdating reveals a temporal TF code in spinal cord neurons

We previously identified sets of TFs that are expressed in multiple subsets of neurons in the spinal cord. As the onset of expression of these TFs occurred at different times during development, we speculated that they subdivide neurons in the spinal cord based on their time point of birth [33] (Fig 1A). We and others have demonstrated before that Onecut TFs are expressed in early-born neurons and that their expression is rapidly extinguished as neurons mature [27,31,33,36,37]. We therefore focused on the TFs Zfhx3, Nfib, and Neurod2, which start to be expressed in neurons at intermediate or late stages during the neurogenic period, respectively, and analyzed the birthdate of neurons expressing these TFs by EdU incorporation (Fig 1B). Pregnant dams were injected with EdU at embryonic day (e)9.5, e10.5, e11.5, or e12.5 (Fig 1C). Embryos were collected at e13.5 and forelimb-level spinal cord cryosections assayed for colocalization between EdU and Zfhx3, Nfib, and Neurod2 in neurons (Figs 1D–1F and S1).

Consistent with the hypothesis of a temporal TF code, a high proportion of EdU-labeled neurons expressed Zfhx3, when EdU was administered at e9.5 and e10.5, while there was little, if any, colocalization between EdU and Zfhx3 when EdU was given at later time points (Figs 1G and S1A). By contrast, few EdU-positive neurons expressed Nfib when EdU was administered before e11.5, but more than 80% of EdU-positive neurons were positive for Nfib when it was given at e12.5 (Figs 1G and S1B). Neurod2 followed a similar trend to Nfib until e11.5 (Figs 1G and S1C), consistent with the high degree of coexpression between these genes [33]. However, the proportion of Neurod2-positive neurons decreased when EdU was given at e12.5 (Figs 1G and S1C). This may be due to the relatively late onset of Neurod2 expression

after neuronal differentiation. Furthermore, Neurod2 is not expressed in late-born dorsal excitatory neurons [40], which are generated at high frequency during late neurogenic stages in the spinal cord [54].

The mutually exclusive birthdates of Zfhx3 and Nfib/Neurod2-positive neurons indicate that these TFs label largely nonoverlapping subsets of neurons. To test this prediction directly, we stained e13.5 spinal cord sections for either Zfhx3 and Nfib or Zfhx3 and Neurod2 (S2A and S2B Fig). Although each of these markers labeled a large number of neurons, the expression of Zfhx3 and Nfib or Zfhx3 and Neurod2 was mutually exclusive. These results are consistent with a model in which Zfhx3 is specifically expressed and maintained in neurons born before e11.5 but not in later-born neurons, which instead express Neurod2/6 and Nfi-family TFs. Together, the data argue against sequential expression of these TFs during neuronal maturation because, in such a model, TFs with an early onset of expression would be specific for early maturation stages and would thus, contrary to our observations, expected to be labeled by EdU given at late developmental time points. Consistent with this interpretation, a recent study found similar birthdates for Zfhx3- and Neurod2-positive neurons in the perinatal spinal cord [41]. We therefore conclude that these TFs comprise a temporal code and label distinct subsets of neurons based on their time point of birth in the spinal cord.

## Conservation of the temporal TF code in the retina

Similar to the spinal cord, Pou2f2, Neurod2/6, and TFs of the Onecut and Nfi families are required for the generation of early and late-born neurons in the retina [42–44,55,56]. We therefore speculated that the temporal TF code is preserved in the retina. To test this hypothesis, we analyzed a published single-cell RNA sequencing (scRNAseq) time course of mouse retina development [43] (S3 Fig). Performing dimensionality reduction by Uniform Manifold Approximation and Projection (UMAP) from prenatal and perinatal stages (e14, e16, e18, P0) resulted in clear trajectories from retinal progenitors to horizontal cells, amacrine cells, retinal ganglion cells, and cone and rod photoreceptors (S3A and S3B Fig). Examining *Onecut2*, *Pou2f2*, *Zfhx3*, and *Nfib* revealed different expression of these genes along these differentiation trajectories (S3C Fig). As expected, *Onecut2* was strongly enriched in horizontal cells, an early-born cell type in the retina, although some expression was also observed in retinal ganglion cells, amacrine cells, and cones. *Nfib* expression was largely restricted to late progenitors and rods (S3C Fig). By contrast, *Pou2f2* and *Zfhx3* were enriched in amacrine and retinal ganglion cells. Furthermore, both genes were expressed in subsets of retinal progenitors.

To further characterize the expression of *Onecut2*, *Pou2f2*, *Zfhx3*, and *Nfib* genes in retinal neurons, we plotted their expression levels in the individual classes of neurons stratified by developmental stage (S3D Fig). This analysis revealed a clear link between the expression of these TFs and developmental stage. *Onecut2* was enriched in amacrine cells, retinal ganglion cells, and cones at e14 (S3D Fig). *Zfhx3* was absent at this stage but was enriched in amacrine and retinal ganglion cells at e18 and P0 (S3D Fig). *Pou2f2* and *Zfhx3* were not detected in cone and rod photoreceptors at any stage, suggesting that not all aspects of this temporal program apply to all neuronal subtypes (S3C and S3D Fig). These data support the hypothesis that the temporal TFs are expressed in different retinal cell types born at distinct time points and raise the possibility that the expression of these genes further subdivide distinct classes of retinal neurons based on their birthdates.

## Conservation of the temporal TF code along the neuraxis

Nfi TFs are expressed in neurons in the forebrain including the cortex, thalamus, and hippo-campus [46,57,58], while Zfhx3 has been implicated in controlling circadian function of the

suprachiasmatic nucleus [59]. Moreover, Pou2f2, Zfhx3, and Nfi TFs are expressed in subpopulations of neurons born from the midbrain floorplate [45]. These results raise the possibility that the sequential expression of the temporal TFs might be broadly preserved throughout the developing nervous system. To test this, we first turned our attention to available scRNAseq time course data from the developing forebrain, midbrain, and hindbrain [9]. Characterization using UMAP dimensionality reduction revealed widespread expression of the temporal TFs in excitatory and inhibitory neurons, identified based on *Slc17a6* (also known as *vGlut2*) and *Gad2* expression, respectively, in the different regions of the nervous system (S4A–S4H Fig). Plotting the dynamics of *Onecut1-3*, *Pou2f2*, *Zfhx3/4*, *Nfia/b*, and *Neurod2/6* in neurons between e8.5 and e14 revealed a striking conservation of the expression dynamics of these TFs. Expression of *Onecut* family TFs preceded *Pou2f2* and *Zfhx3/4*, while *Nfia/b* and *Neurod2/6* were only expressed at high levels at later stages (Fig 2A). Consistent with this observation, hierarchical clustering on the expression patterns of these TFs revealed the same correlation patterns as in the spinal cord (S4I–S4L Fig). Moreover, ranking gene expression patterns by correlation to *Nfib* or *Zfhx3* expression revealed that the TFs that define the same temporal identity windows are typically among the best-correlated genes in each region (e.g., *Nfia*, *Nfix*, *Tcf4*, *Neurod2*, and *Neurod6* for *Nfib* or *Pou2f2*, *Zfhx4*, and *Zfhx2* for *Zfhx3*), while the TFs that define different temporal identity windows typically rank among the most anticorrelated genes genome-wide (S4M Fig). Together, these data suggest that the same temporal patterning program identified in the spinal cord applies to large regions of the developing central nervous system.

To validate experimentally these predictions, we turned to immunofluorescent analysis of Onecut2, Pou2f2, Zfhx3, and Nfib in hindbrain and midbrain cryosections from different developmental stages (Fig 2B–2H). In both tissues, the majority of neurons expressed Onecut2, but not Pou2f2, at early developmental stages (e9.5 or e10.5, respectively), and both genes were expressed in largely nonoverlapping populations of neurons 1 day later (Fig 2B, 2E and 2F). Furthermore, in both tissues, a large proportion of neurons expressed Zfhx3 at e11.5, while Nfib expression was confined to neural progenitors at this stage (Fig 2C, 2G and 2H). At e13.5, Nfib-positive cells, which had lost the expression of the progenitor marker Sox2, were detected in the mantle layer of both tissues where postmitotic neurons reside (Figs 2C, 2G, 2H and S5A). These cells did not express Sox9, suggesting that they were not glial progenitors (S5B Fig). To test if these Nfib-expressing cells are neurons, we costained hindbrain sections for Phox2b, which is expressed in different populations of hindbrain neurons [60]. This analysis revealed colocalization between Phox2b/Zfhx3 and Phox2b/Nfib in individual nuclei (Fig 2D), suggesting that Nfib indeed labels late-born neurons in the hindbrain. Similarly, many Nfib-positive nuclei in the mantle layer were also positive for another neuronal marker, Lhx5, further confirming their neuronal identity (S5C Fig). These results demonstrate that the temporal TFs are expressed in largely nonoverlapping populations of neurons throughout the developing nervous system. Furthermore, their expression commences in the same sequence (Onecut2 -> Pou2f2/Zfhx3 -> Nfia/Nfib) as in the spinal cord [33], consistent with the idea that these TFs are expressed in neurons generated at different time points during embryonic development.

To further test this hypothesis, we used scRNAseq time course data to reconstruct gene expression dynamics of neuronal differentiation at different time points of embryonic development (S6A–S6C Fig and S1 Text). We reasoned that if neurons expressing the temporal TFs are generated sequentially during embryonic development, they should be at different stages of their maturation when cells were collected for scRNAseq (S6A Fig). Earlier-born neurons should thus occupy later pseudotime bins in the neuronal differentiation trajectory than recently generated neurons. Hence, the expression of the temporal TFs should appear as

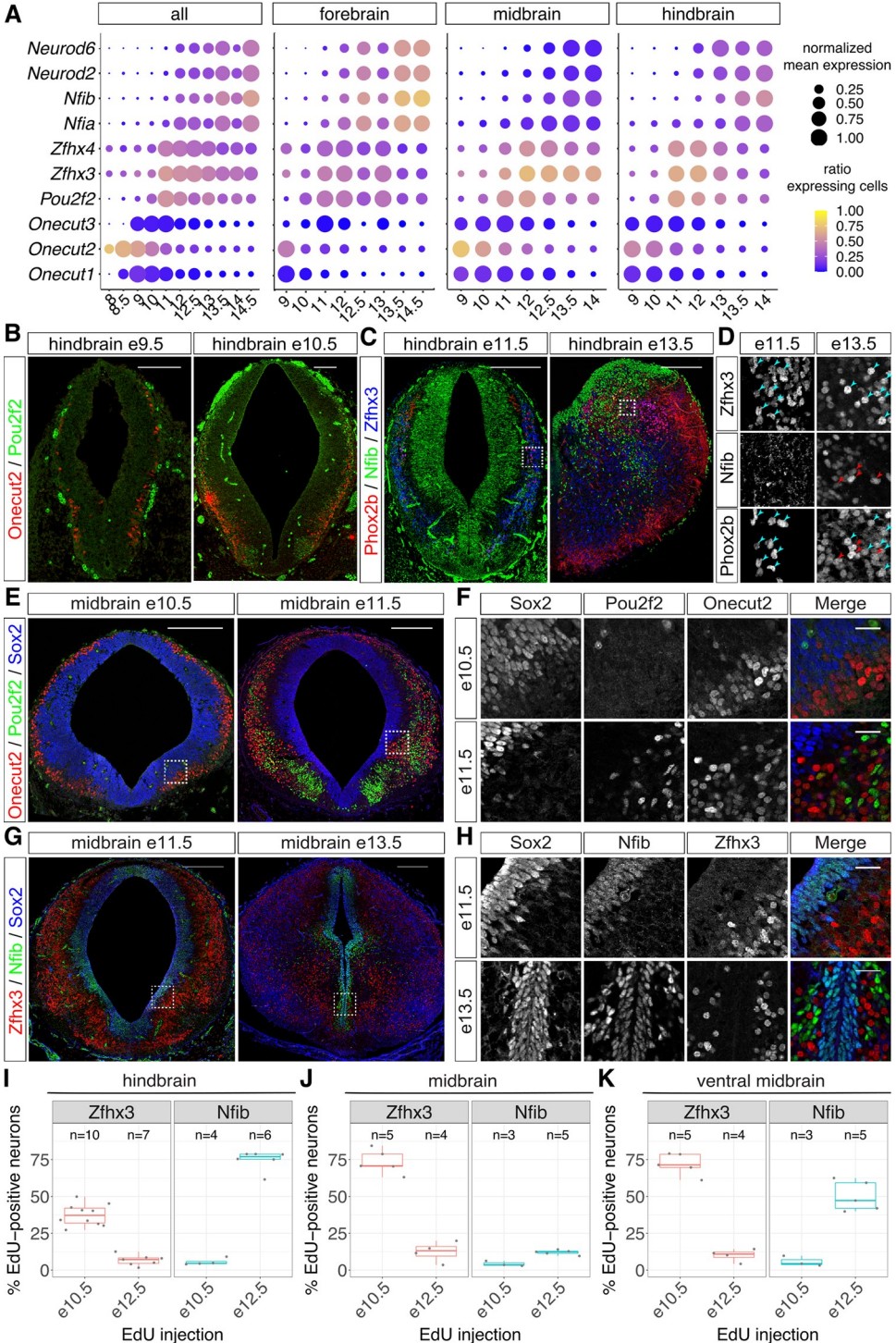

**Fig 2. The temporal TF code is conserved at different rostral–caudal levels of the nervous system (see also S3–S9 Figs).** (A) Expression of temporal TFs in scRNAseq data [9] from the developing forebrain, midbrain, and hindbrain suggests conservation of temporal patterning in these parts of the nervous system. (B-D) Conservation of temporal patterning in the hindbrain. (B) Onecut2, but not Pou2f2, is expressed in hindbrain neurons at e9.5, while both TFs label distinct populations of neurons at e10.5. (C) Zfhx3, but not Nfib, labels neurons at e11.5. These TFs label distinct populations of neurons at e13.5. (D) Zfhx3 and Nfib label distinct subsets of Phox2b-positive neurons in the hindbrain at e13.5. (E-H) Conservation of temporal patterning in the midbrain. (F, H) show higher magnification images of the regions outlined in (E, G), respectively. (E, F) Onecut2, but not Pou2f2, labels neurons in the midbrain at e10.5. Both

TFs label distinct subsets of neurons at e11.5. (G, H) Zfhx3 labels neurons at e11.5, while Nfib expression is restricted to neural progenitors. At e13.5, Zfhx3 and Nfib label distinct subsets of neurons in the midbrain at e13.5. (I-K) Percentage of EdU-positive neurons labeled by Zfhx3 and Nfib in the hindbrain (I), entire midbrain (J), and ventral and intermediate midbrain only (K). Underlying data are provided in S2 Data. Scale bars = 100 μm (B), 200 μm (C, E, G), or 25 μm (D, F, H). scRNAseq, single-cell RNA sequencing; TF, transcription factor.

largely nonoverlapping waves moving in pseudotime (S6B and S6C Fig). To test this approach, we first focused on the spinal cord, where experimental evidence indicates that the temporal TFs are expressed in neurons born at different stages (Figs 1 and S1). Reconstructing the pseudotemporal dynamics of several neuronal lineages along the dorsal–ventral axis, such as dI1, dI5, V2a, and V2b neurons revealed that the expression profiles of the temporal TFs indeed behaved as expected (S6D Fig). *Zfhx3/4* consistently appeared before *Nfia/b* and *Neurod2/6* expression and occupied later pseudotime bins at later developmental stages (S6D Fig). These observations are consistent with the successive generation of these neurons and argue against sequential expression of these TFs in the same neurons.

We next focused on the hindbrain (S6E Fig), which shares many neuronal populations with the spinal cord. Reconstructing the expression dynamics of large groups of neurons, such as ventral Pax2-positive inhibitory interneurons, or of specific neuronal lineages, such as V1, dB4, or Hox-negative dA1 neurons, revealed similar pseudotemporal expression dynamics of the temporal TFs (S6E Fig). Finally, we asked whether similar expression dynamics were observed in neuronal populations in the forebrain. Performing pseudotemporal ordering of cortical excitatory and inhibitory neurons revealed the same sequential expression patterns of the temporal TFs in these neuronal populations as in other regions of the neuraxis (S6F Fig). The same signature of temporal TF expression was also recovered when pseudotemporal expression dynamics were reconstructed for defined groups of inhibitory interneurons, such as *Lhx6*-expressing neurons derived from the medial ganglionic eminence or *Meis2/Isl1/Ebf1*-expressing neurons derived from the lateral ganglionic eminence (S6F Fig). These data provide further evidence that the temporal TFs are expressed in neurons born at different time points in a wide range of neuronal subtypes throughout the developing nervous system.

## EdU birthdating confirms sequential generation of Zfhx3 and Nfib-positive neurons

We next sought to confirm experimentally that Zfhx3 and Nfib-positive neurons are generated at distinct time points in the midbrain and hindbrain using EdU birthdating. To this end, we injected pregnant dams at e10.5 or e12.5 with EdU and assayed the proportion of EdU-positive neurons expressing Zfhx3 or Nfib at e13.5 in the midbrain or hindbrain. As for the spinal cord (Fig 1), Zfhx3-positive neurons in both tissues were labeled by EdU at e10.5 but not at e12.5 (Figs 2I and 2J and S7A–S7D). Moreover, in the hindbrain, a high proportion of EdU-positive neurons expressed Nfib when EdU was given at e12.5 (Figs 2I, S7E, and S7F). By contrast, the proportion of Nfib-positive neurons in the midbrain did not increase markedly (Fig 2J). This was because most neurons labeled by EdU at e12.5 resided in the dorsal part of the midbrain where Nfib is not expressed in neurons at this stage (S7G and S7H Fig). These neurons did not express Zfhx3 (S7D Fig), suggesting that the lack of Nfib-positive neurons in this area was not due to prolonged generation of Zfhx3 neurons. Restricting the analysis to the intermediate and ventral midbrain resulted in the expected increase of EdU-positive neurons expressing Nfib (Fig 2K). Taken together, these observations provide further experimental evidence that Zfhx3 and Nfib label sequentially generated neurons in large regions of the midbrain and hindbrain.

## Most cortical excitatory neurons express late temporal TFs

Molecularly and functionally distinct excitatory neurons in the mammalian cortex are arranged in distinct layers based on their time point of generation [18,20,61]. As this is one of the best-established models for temporal neuronal subtype generation, we wondered how the temporal patterning program we describe relates to the temporal patterning of the cortical layers. To this end, we analyzed scRNAseq of cortical excitatory neurons from e10 to e14 [9] (S8A and S8B Fig). Consistent with the extended period of neurogenesis in the cortex compared to other regions of the nervous system, we found that most cortical excitatory neurons express TFs characteristic of the late temporal identity, including *Nfia*, *Nfib*, *Neurod2*, *Neurod6*, and *Tcf4* (S8B and S8C Fig). However, a small cluster (cluster 7; S8D Fig) of excitatory neurons lacked expression of these markers. Instead, these neurons expressed *Pbx3*, *Meis1*, *Meis2*, *Tshz2*, *Barhl2*, and *Zfhx3* (S8C and S8E Fig). Consistent with the timing of generation of Zfhx3 neurons in the rest of the nervous system, the cortical Zfhx3-positive neurons appear to be generated earlier during development than Nfia/b Neurod2/6-positive neurons (S8F Fig). Taken together, these data suggest that the temporal patterning program we describe applies to cortical neurons and that most excitatory cortical neurons express TFs characteristic of a late temporal identity. These findings match recent observations by Moreau and colleagues, demonstrating a subdivision of early cortical excitatory neurons into Pbx3/Zfhx3 and Nfi/Neurod2/Neurod6-positive subtypes [46].

## Conservation of temporal TF expression at later developmental stages

Zfhx3 and Neurod2/Nfib also partition spinal cord neurons in the perinatal and adult spinal cord [41]. We therefore investigated if the subdivision into Zfhx3 and Nfib-positive neurons is maintained in other regions of the nervous system. To address this, we analyzed scRNAseq data from late embryonic (e16 to e18) forebrain and midbrain [9]. Characterization of the gene expression patterns of intermediate (*Pou2f2*, *Zfhx3*, *Zfhx4*) and late (*Nfia*, *Nfib*, *Neurod2*, *Neurod6*, *Tcf4*) TFs indicated that these TFs continue to be expressed in largely nonoverlapping populations of neurons (S9A, S9B, S9F, and S9G Fig). *Nfia/Nfib*-positive cells expressed the neuronal markers *Elavl3* and *Tubb3* but did not express the glial markers *S100b* and *Slc1a3* (also known as *Glast*), confirming their neuronal identity (S9C and S9H Fig). Moreover, hierarchical clustering of the gene expression patterns and correlation rank plots for *Zfhx3* and *Nfib* revealed similar coexpression patterns as at earlier developmental stages (S9D, S9E, S9I, and S9J Fig). Collectively, these results suggest that the anticorrelated expression of intermediate and late TFs are broadly retained until late embryonic stages in the forebrain and midbrain.

## Temporal TF expression correlates with the acquisition of distinct neuronal identities in the ventral midbrain

We next investigated whether the temporal TF code is responsible for the establishment of neuronal populations with specific functions. To test this, we first focused on the sequential generation of oculomotor and red nucleus neurons in the ventral midbrain. The switch from oculomotor to red nucleus neurons occurs between e10.5 and e11.5 [22,48]. We therefore speculated that this switch may coincide with the switch from early (Onecut2) to intermediate (Zfhx3) TF expression. Consistent with this hypothesis, expression of the red nucleus neuron marker Pou4f1 is mutually exclusive with the expression of Onecut2 in the ventral midbrain (S10A Fig), while most Pou4f1 neurons express Zfhx3 (S10B Fig). These data support the hypothesis that the temporal TFs are involved in the sequential generation of neurons with

distinct functions. Of note, Pou4f1 and Onecut2 are coexpressed in subsets of neurons in the dorsal midbrain, suggesting that these TFs do not always mutually cross-repress each other.

Dopaminergic neurons are a neuronal population of medical interest because their degeneration causes Parkinson disease. During development, these neurons are born from the midbrain floor plate and can be discriminated based on the expression of the TFs Lmx1a, Lmx1b, and Pitx3 as well as the enzymes tyrosine hydroxylase (TH) and the dopamine transporter Slc6a3 (also known as Dat). Strikingly, previous characterization of neurons generated from the midbrain floor plate suggested that these neurons can be broadly subdivided into Nfia/b and Zfhx3 expressing subsets. The Zfhx3-positive population expresses dopaminergic neuron markers such as Slc6a3 and high levels of TH [45]. By contrast, the Nfi-positive population lacked the molecular machinery for the synthesis of dopamine and expressed markers characteristic for excitatory neurons such as Slc17a6 [45]. These findings, in combination with our observation that Zfhx3 and Nfi TFs define temporal neuronal populations in the midbrain, suggest that midbrain dopaminergic neurons may constitute a temporal neuronal subtype born from the midbrain floor plate.

We therefore examined if Zfhx3-positive neurons are generated before Nfia/b-positive neurons from the midbrain floor plate. Assays at e11.5 revealed widespread expression of Zfhx3 in floor plate–derived Lmx1b-positive neurons (Fig 3A). At this stage, Nfib expression just commenced in Sox2-positive neural progenitors (Fig 3B). In contrast, at e13.5, numerous Nfib-positive neurons expressing Lmx1b were found in the vicinity of the midbrain floor plate (Fig 3C and 3D), likely corresponding to the $N\text{-}Dat^{low}$ population [45]. Zfhx3-positive neurons at this stage had migrated to a more lateral position (Fig 3C). These neurons coexpressed the Zfhx TFs, Zfhx3, and Zfhx4 and also increased levels of TH (Fig 3E and 3F), suggesting that these populations correspond to the $AT\text{-}Dat^{high}$, $T\text{-}Dat^{high}$, and $VT\text{-}Dat^{high}$ neurons described by Tiklová and colleagues. These conclusions are also consistent with previous birthdating experiments that concluded that the majority of TH-positive dopaminergic neurons are born before and around e12.5 [62,63]. Taken together, these data suggest that the sequence of temporal TF expression is preserved in neurons derived from the midbrain floor plate, that the expression of different temporal TFs correlates with the acquisition of different neuronal subtype identities in these neurons, and that dopaminergic neurons correspond to the Zfhx3-positive temporal neuronal population.

## The temporal TF code applies to in vitro generated midbrain, hindbrain, and spinal cord neurons

We next sought to investigate whether the temporal code was preserved in vitro during the directed differentiation of embryonic stem (ES) cells to neurons with specific axial and dorsal–ventral identities [64–66]. We reasoned that in vitro putative global signaling cues, originating from distant signaling centers, should be absent.

We examined if the same sequence of temporal TF factor expression can be observed in stem cell–derived neurons with midbrain and hindbrain and spinal cord identities. ES cells were differentiated to appropriate identities using established protocols [64] (Fig 4A), as confirmed by real-time quantitative polymerase chain reaction (RT-qPCR) for *Foxg1*, *Otx2*, *Hoxa4*, *Hoxb9*, and *Hoxc8* (S11A Fig). As expected, cells differentiated to midbrain identity induced *Otx2*, but not the forebrain marker *Foxg1* or the hindbrain marker *Hoxa4*, which was induced in hindbrain conditions. By contrast, the posterior Hox genes *Hoxb9* and *Hoxc8* were only induced when cells were differentiated to a spinal cord identity. We next assayed the expression of the temporal TFs Onecut2, Zfhx3, Nfia, and Neurod2 under these differentiation conditions by flow cytometry from days 6 to 13 (Figs 4B and S11B). The overall expression

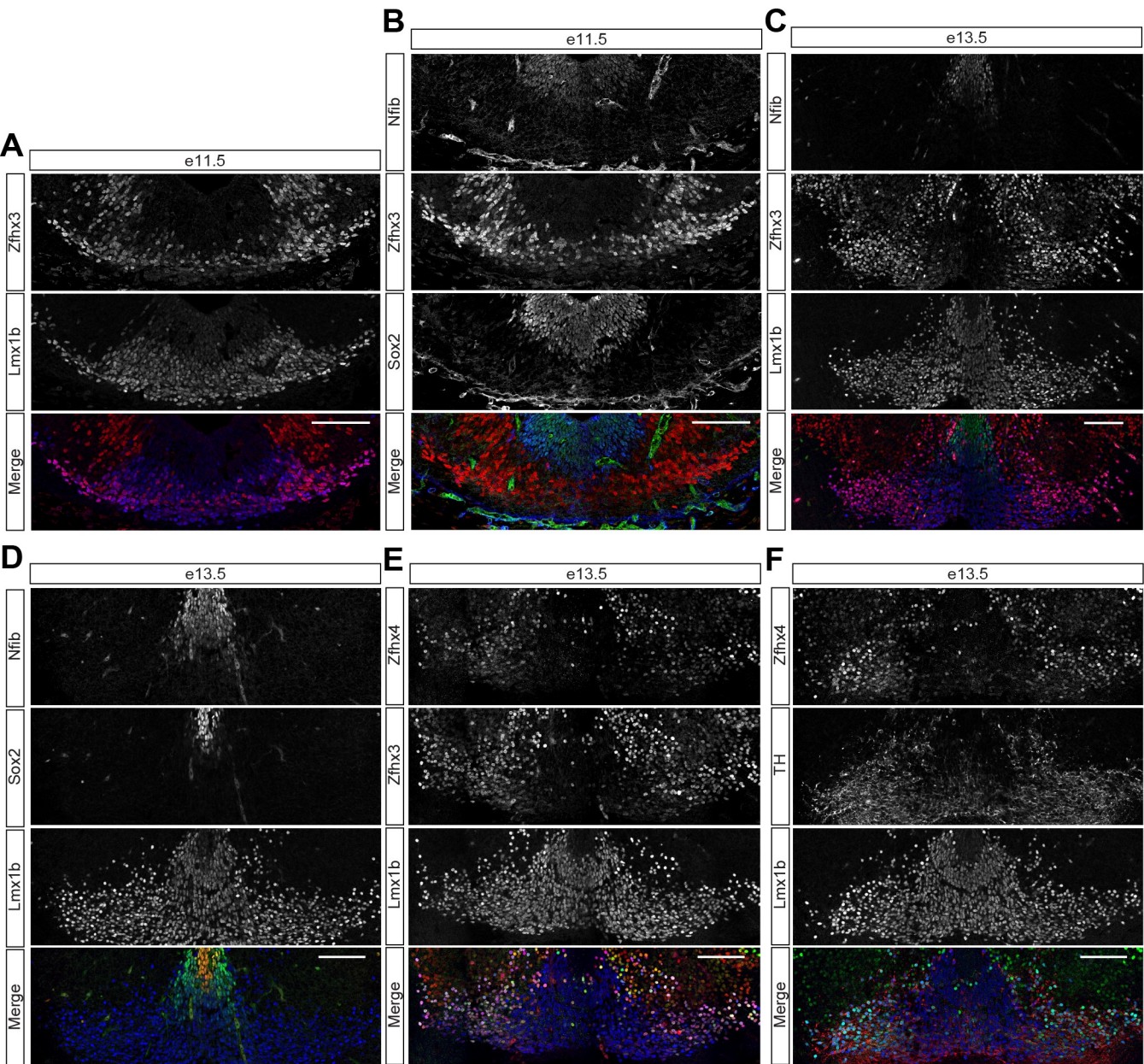

**Fig 3. Midbrain dopaminergic neurons are a temporal population of neurons derived from the midbrain floor plate (see also S10 Fig).** (A) Coexpression of Zfhx3 and Lmx1b in neurons derived from the midbrain floor plate at e11.5. (B) Nfib is restricted to Sox2-positive neural progenitors in the ventral midbrain at e11.5. (C) Mutually exclusive expression of Zfhx3 and Nfib in Lmx1b-positive neurons at e13.5. (D) Nfib labels Lmx1b-positive neurons directly adjacent to Sox2-positive progenitors at e13.5. (E) Colocalization between Zfhx3 and Zfhx4 in Lmx1b-positive neurons at e13.5. (F) Zfhx4 labels Lmx1b-positive neurons expressing high levels of TH at e13.5. Scale bars = 100 μm. TH, Tyrosine hydroxylase.

dynamics of these markers observed in vivo were preserved under the different conditions. Most neurons expressed Onecut2 at days 6 and 7, while the proportion of Zfhx3-positive neurons increased between days 7 and 9, and Nfia and Neurod2-positive neurons were typically not detected before day 11. These results closely resemble our previous observations of the temporal patterning of neurons in the developing nervous system.

We next investigated if the progression of the temporal TF code is preserved in neurons with different dorsal–ventral identities. We have previously demonstrated that exposure of

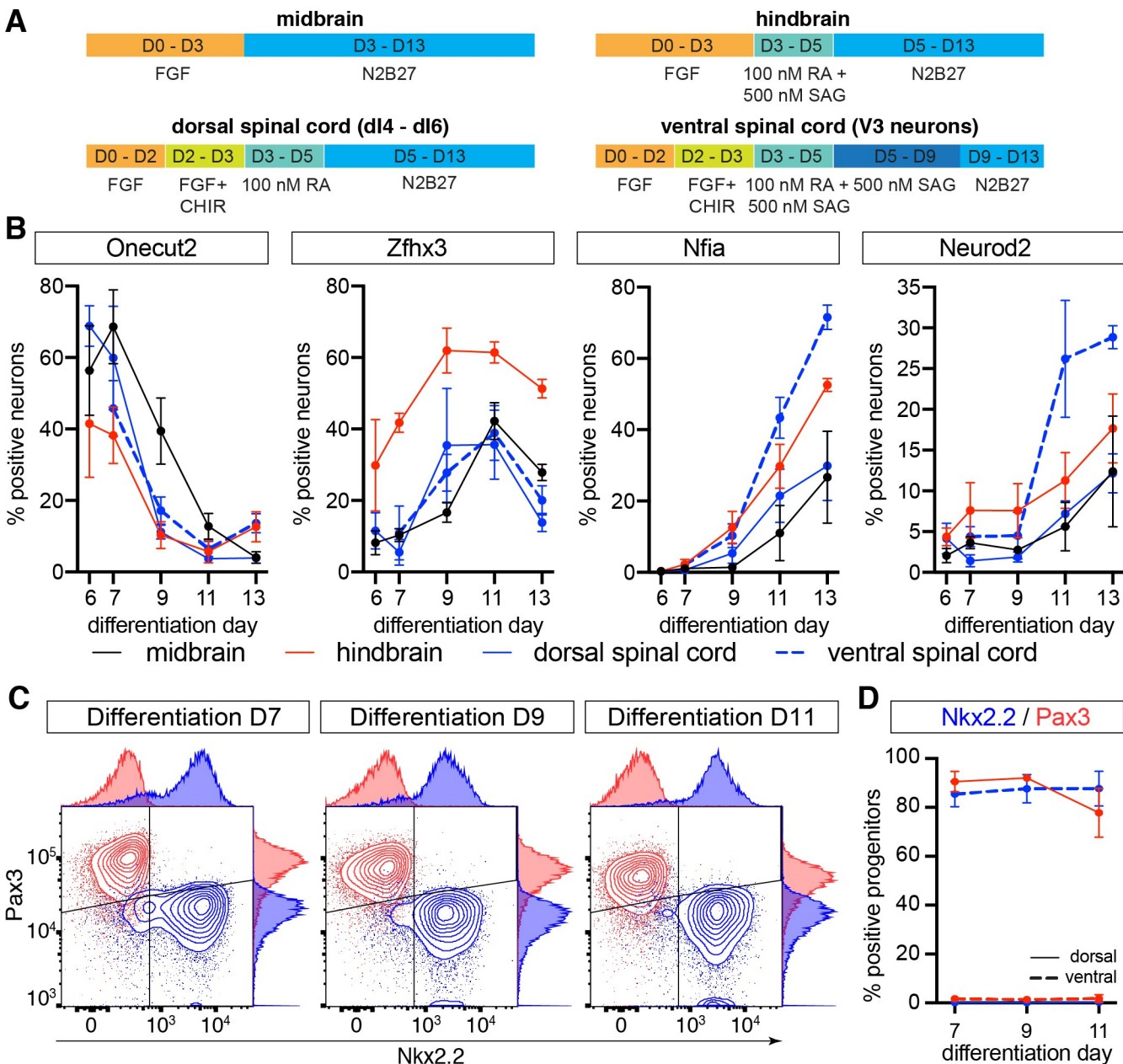

**Fig 4. Conservation of the temporal TF code in stem cell–derived neurons with different axial and dorsal–ventral identities (see also S11 Fig).** (A) Schematics of the differentiation protocols for the generation of progenitors and neurons with different axial and dorsal–ventral identities. (B) Flow cytometry analysis of temporal TF expression indicates that neurons with different axial and dorsal–ventral identities display the same temporal progression in vitro as in vivo. (C) Flow cytometry analysis of Nkx2.2 and Pax3 expression in neural progenitors in dorsal and ventral spinal cord differentiations. (D) Percentage of neural progenitors expressing Pax3 and Nkx2.2 in ventral and dorsal spinal cord differentiations between days 7–11. Underlying data are provided in S3 Data. FGF, fibroblast growth factor; RA, retinoic acid; SAG, Shh pathway agonist; TF, transcription factor.

spinal cord progenitors to appropriate concentrations of the Sonic Hedgehog (Shh) pathway agonist (SAG) promotes the generation of progenitors and neurons with different dorsal–ventral identities [66]. We therefore focused on the spinal cord condition and either ventralized cells by exposing them from day 3 to day 9 to 500 nM SAG or dorsalised them in the absence of SAG. Samples for flow cytometry were collected at days 7, 9, and 11 (Fig 4C). Consistent with our previous observations [66], in the absence of Shh pathway activation, most

progenitors expressed the dorsal progenitor marker Pax3, while prolonged high-level Shh pathway activation leads to the majority of progenitors acquiring an Nkx2.2-positive ventral p3 identity (Fig 4C and 4D). Consistent with this, most neurons generated in the absence of Shh pathway activation expressed the intermediate dorsal marker Lbx1, while Shh pathway activation led to the generation of Sim1-positive V3 neurons (S11C Fig). We therefore refer to these conditions as dorsal and ventral, respectively. Assaying the expression of the temporal TFs in neurons in the ventral differentiation condition revealed similar expression dynamics for these markers as previously observed under dorsal spinal cord conditions, although notably a higher proportion of neurons expressed Nfia and Neurod2 at later stages of the differentiations (Fig 4B).

Based on these oberservations, we conclude that the temporal TF code is preserved in in vitro generated neurons with different axial (midbrain, hindbrain, and spinal cord) and dorsal–ventral identities. Furthermore, the time scale over which the temporal patterning unfolds is similar in vivo and in vitro, corresponding in both cases to approximately 4 to 5 days (in vivo approximately e9.5 to e13.5; in vitro approximately day 7 to day 11). These results argue against a model in which global signaling cues orchestrate the temporal patterning program. We note, however, that this analysis also uncovered reproducible differences in the proportions of neurons expressing the respective markers between the different axial identities. Cells differentiated under hindbrain conditions induced late temporal TFs at a faster pace, while cells under midbrain conditions seemed to progress slowest to a later temporal identity. These differences may be indicative of cell-intrinsic programs that allow progenitors and/or neurons to progress through the temporal TF code at a speed characteristic for their axial identity.

## Conserved temporal patterning of midbrain, hindbrain, and spinal cord neural progenitors

Temporal neuronal subtype specification is arguably best understood in *Drosophila*. Here, aging neuroblasts sequentially express a series of TFs that define temporal identity windows for the generation of specific neuronal progeny [12–14]. Similar processes are believed to underlie the temporal patterning of tissues in the vertebrate nervous system; however, the transcriptional programs that mediate this process are still relatively poorly understood [11,17]. We therefore asked if similar principles apply to the spinal cord. To this end, we analyzed our in vivo scRNAseq data [33] to identify TFs that are consistently up- or down-regulated in most progenitor domains during the neurogenic period (see Experimental procedures). This analysis recovered in total 542 genes including 33 TFs (Fig 5A). Inspection of the expression dynamics of these TFs confirmed their differential temporal expression in progenitors from most dorsal–ventral domains (S12A Fig). As expected, this analysis recovered the gliogenic TF *Sox9* and the Nfi TFs (*Nfia/b/x*) that have previously been shown to be dynamically expressed during this time window in the developing spinal cord [67,68]. To address if these transcriptional changes are preserved in progenitors in other regions of the nervous system, we characterized the expression dynamics of the 33 TFs in scRNAseq from the developing forebrain, midbrain, and hindbrain [9]. This analysis revealed largely preserved expression dynamics of the 33 TFs in these tissues (Fig 5B).

We next tested if these 33 TFs display the same expression dynamics in neural progenitors in our in vitro differentiations. Analysis of the gene expression dynamics of the 542 genes and 33 TFs using RNAseq data from in vitro generated ventral neural progenitors from days 3 to 10 [69] revealed that the general temporal pattern of gene expression is preserved under these culture conditions (Fig 5C). To better characterize the differences in gene expression between in vivo and in vitro, we partitioned the 542 genes into correlated (Pearson correlation between in vivo and in vitro > 0.5), uncorrelated (correlation between −0.5 and 0.5) and anticorrelated genes (correlation < −0.5) (see Experimental procedures and S5 Data). About 431 genes and 28 TFs showed

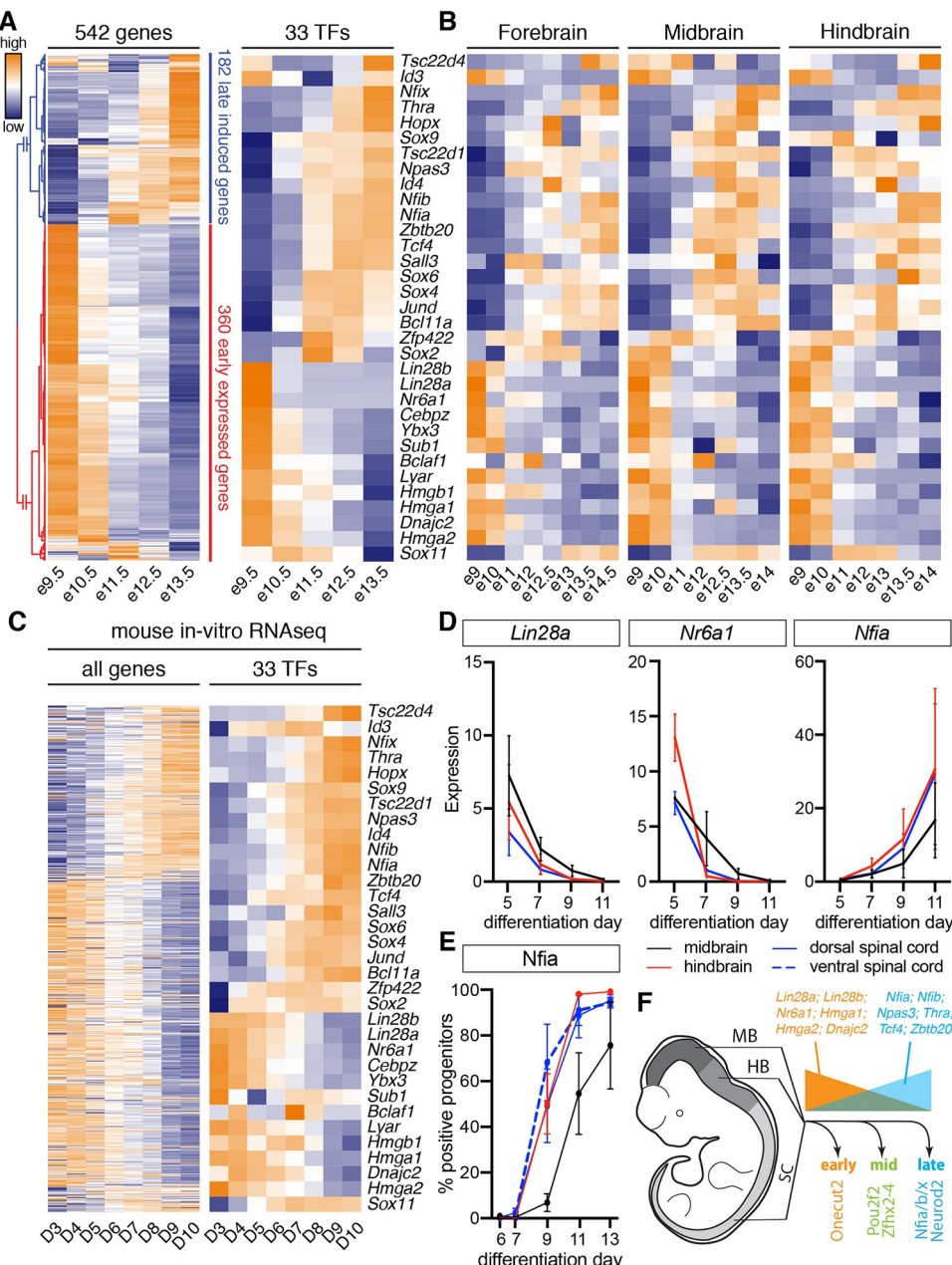

**Fig 5. Conserved temporal patterning of neural progenitors throughout the developing central nervous system (see also S12 and S13 Figs).** (A) Differential gene expression analysis using scRNAseq from spinal cord neural progenitors [33] identifies 542 genes (left) including 33 TFs (right) that are differentially expressed during the neurogenic period. Heatmap shows log-scaled and z-scored gene expression values for each gene. (B) Characterization of the expression dynamics of the same 33 TFs in scRNAseq from the developing forebrain, midbrain, and hindbrain [9]. (C) Expression dynamics of the 542 genes (left) and 33 TFs (right) in RNAseq data from ventral spinal cord differentiations [69]. Heatmap shows log-scaled and z-scored gene expression values for each gene. Order of the genes in both heatmaps is the same as in (A). (D) RT-qPCR analysis of *Lin28a*, *Nr6a1*, and *Nfia* from days 5–11 in in vitro differentiations with different axial identities reveals conserved expression dynamics of these markers in the in vitro differentiations. See S13D Fig for quantification of further markers. Underlying data are included in S4 Data. (E) Quantification of Nfia induction in in vitro generated neural progenitors with different axial identities by flow cytometry. For underlying data, see S3 Data. (F) Conserved temporal patterning of neural progenitors throughout the developing nervous system. Early neural progenitors express markers such as *Lin28a*, *Lin28b*, *Nr6a1*, *Hmga1*, *Hmga2*, and *Dnajc2* (orange), while late progenitors are characterized by the expression of *Nfia*, *Nfib*, *Npas3*, *Thra*, *Tcf4*, and *Zbtb20* (light blue). HB, hindbrain; MB, midbrain; RT-qPCR, real-time quantitative polymerase chain reaction; SC, spinal cord; scRNAseq, single-cell RNA sequencing; TF, transcription factor.

correlated expression dynamics between in vivo and in vitro (S13A Fig). Of the remaining genes, 74 genes including 3 TFs (*Sox11*, *Sox2*, and *Bclaf*) showed an uncorrelated pattern (S13B Fig), while the expression of 36 genes and 2 TFs (*Id3* and *Sub1*) was anticorrelated (S13C Fig).

To test if the same dynamics are also observed in in vitro generated progenitors with mid-brain, hindbrain, or dorsal spinal cord identities, we performed RT-qPCRs for *Lin28a*, *Lin28b*, *Nr6a1*, *Sox9*, *Npas3*, *Zbtb20*, *Nfia*, *Nfib*, and *Hopx* and quantified the proportion of Nfia-positive progenitors by flow cytometry (Figs 5D, 5E, and S13D). These results confirmed shared expression dynamics for these marker genes in in vitro differentiated neural progenitors with different axial identities. We conclude that, similar to neurons, neural progenitors throughout the nervous system undergo a shared temporal patterning program (Fig 5F).

## TGFβ controls the pace of the temporal program

The TGFβ signaling pathway controls the timing of the switch from MN to serotonergic neuron production in p3 progenitors in the vertebrate hindbrain and promotes Nfia expression and the formation of glia in neural stem cells [48,50]. In the spinal cord, the signaling pathway is active in progenitors during the neurogenic period, and several members of the TGFβ family are expressed at early developmental stages in the adjacent notochord, floor plate, and mesoderm and at later developmental stages by different populations of neurons [53,70,71]. We therefore asked if the pathway is active in progenitors in our in vitro differentiations. To do so, we exposed dorsal neural progenitors from day 5 to the TGFβ signaling inhibitor SB431542 [72] and assayed the expression of the target gene *Smad7* 48 and 96 hours later [73,74]. As expected, pathway inhibition resulted in a significant reduction of *Smad7* expression at day 7, although expression partially recovered by day 9 (Fig 6B). These results confirm that the TGFβ pathway is active in neural progenitors in vitro and suggest that TGFβ signaling is a good candidate to control the maturation of progenitors and the timing of temporal TF expression in in vitro generated spinal cord neurons.

To test this hypothesis, we exposed progenitors under dorsal and ventral spinal cord conditions to SB431542 from day 5 onwards (Fig 6A). This treatment did not result in a change in the proportion of progenitors expressing Pax3 or Nkx2.2, suggesting that it does not strongly affect the dorsal–ventral identity of neural progenitors (Fig 6C) but caused a significant delay in the induction of the late marker Nfia in neural progenitors and the expression of the intermediate and late-born markers Zfhx3, Nfia, and Neurod2 in neurons under dorsal and ventral conditions (Fig 6D and Fig 6E).

To investigate further the consequences of TGFβ pathway inhibition on the temporal patterning of neural progenitors, we additionally assayed the consequences of ectopic TGFβ pathway activation and inhibition on the expression of the early genes *Lin28a*, *Lin28b* and the late genes *Sox9*, *Nfia*, *Nfib*, and *Nfix* by RT-qPCR (Fig 6F). This analysis revealed a faster down-regulation of early progenitor and earlier induction of late progenitor markers upon exposure to 2 ng/ml TGFβ2 ligand (Fig 6G), while the opposite was true when the TGFβ pathway was inhibited using 10 μM SB431542 (Fig 6H). Together, these experiments demonstrate that TGFβ signaling controls the speed of progenitor maturation and the timing of temporal TF expression in in vitro generated spinal cord neurons.

## Nfia and Nfib are required for the efficient generation of late-born spinal cord neurons

Nfi TFs are best known for promoting the switch from neurogenic to gliogenic progenitors [50,68,75,76]. However, the expression of Nfia and Nfib in progenitors in the mouse spinal cord commences between e10.5 and e11.5, approximately 2 days before neurogenesis ceases and gliogenesis starts. Furthermore, these TFs are expressed in different types of neurons

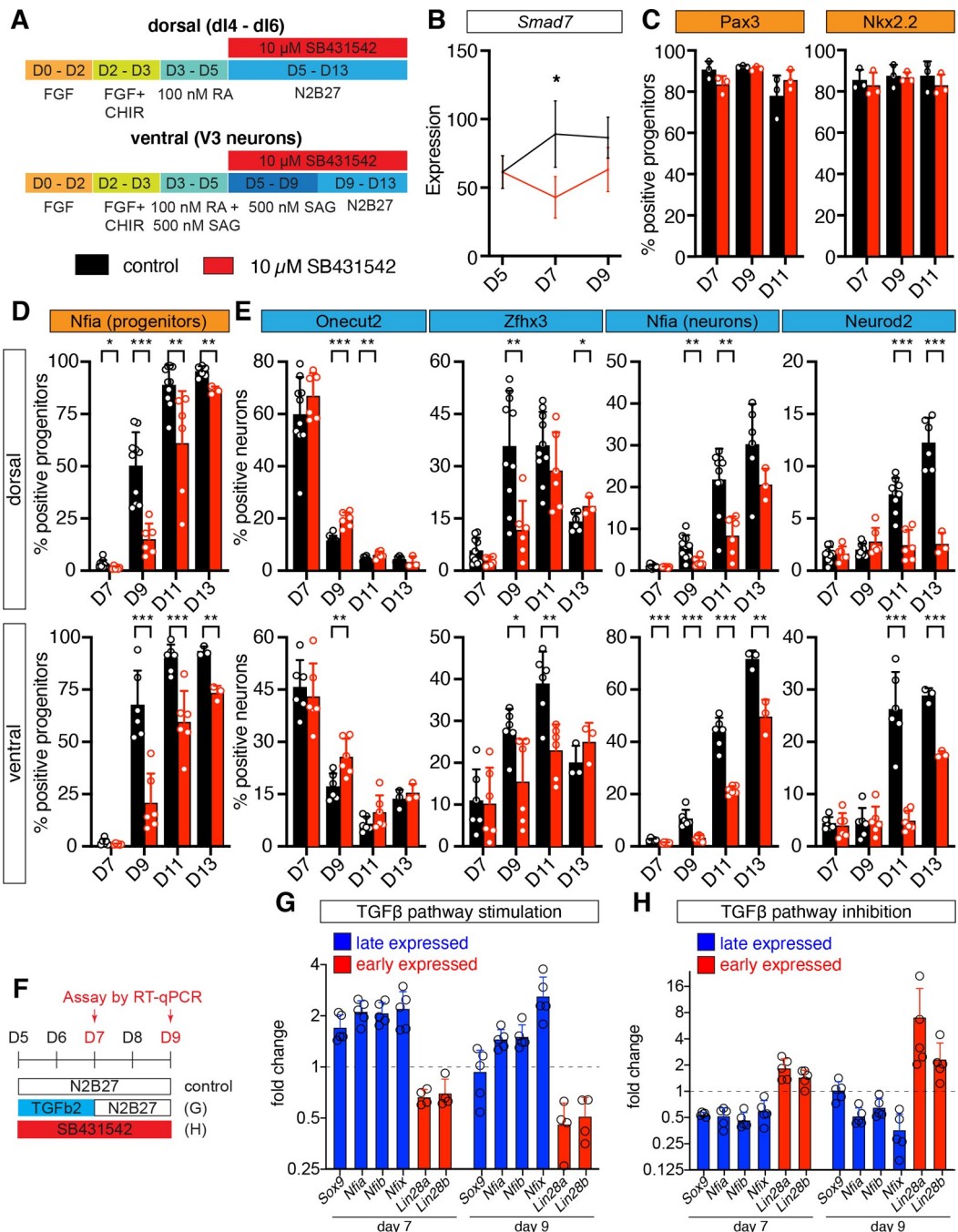

**Fig 6. TGFβ signaling influences the timing of temporal TF expression in neurons and progenitors.** (A) Schematics of the differentiation protocols for TGFβ pathway inhibition in dorsal and ventral spinal cord conditions. (B) Inhibition of TGFβ signaling in dorsal spinal cord conditions causes down-regulation of the TGFβ pathway target gene *Smad7*. (C) TGFβ pathway inhibition does not alter the proportion of progenitors expressing Pax3 in dorsal (left) or Nkx2.2 in ventral (right) conditions. (D) Inhibition of TGFβ signaling delays the induction of Nfia in dorsal and ventral spinal cord neural progenitors. (E) Percentage of neurons expressing the different temporal TFs in the presence and absence of TGFβ pathway inhibition. TGFβ pathway inhibition causes a delay in the induction of the late neuronal markers Zfhx3, Nfia, and Neurod2 in neurons. (F) Scheme outlining the differentiation protocol to assess the role of TGFβ pathway activation and inhibition on the temporal patterning of neural progenitors. (G) TGFβ pathway activation causes an earlier induction of the late markers *Sox9*, *Nfia*, *Nfib*, and *Nfix* and earlier down-regulation of *Lin28a* and *Lin28b* by RT-qPCR. (H) TGFβ pathway inhibition has the opposite effect on the expression of these markers. Underlying flow cytometry data are provided in S3 Data, qPCR data in S4 Data. FGF, fibroblast growth factor; RA, retinoic acid; RT-qPCR, real-time quantitative polymerase chain reaction; SAG, Shh pathway agonist; TF, transcription factor.

throughout the developing nervous system [33,45,57] (S4E–S4H Fig). In the retina, Nfi TFs are required for the specification of Müller glia and, importantly, bipolar cells, a late-born neuronal subtype [43]. These findings raise the possibility that Nfi TFs are also required for the specification of late-born neuronal subtypes in other parts of the central nervous system. To test this possibility, we generated an *Nfia*; *Nfib* double-mutant ES cell line by CRISPR/Cas9-induced nonhomologous end joining. Because Nfia and Nfib act redundantly during the induction of gliogenesis in the spinal cord and the formation of bipolar cells and Müller glia in the retina [43,68], we decided to focus on analyzing the double mutant to rule out any potential redundancy between both genes. Electroporations of guide RNAs targeting the second coding exons of both genes resulted in double-heterozygous frameshift deletions of 20 and 11 base pairs in *Nfia* and 10 and 8 base pairs in *Nfib* (S14A and S14B Fig). Immunofluorescence assays of dorsal differentiations at day 10 of differentiation, a time point when both proteins are normally detected at high levels in progenitor nuclei in control differentiations, confirmed the absence of both proteins (S14C and S14D Fig).

Throughout the developing nervous system, the expression of *Neurod2* is among the most correlated with *Nfia* and *Nfib* expression in neurons (S4I–S4M Fig). Furthermore, both in vivo and in vitro Neurod2-positive neurons are born after Nfia and Nfib expression commenced in progenitors [33] (compare Figs 4B and 5E). Moreover, analysis of Nfia and Nfib ChIP-seq data from the postnatal murine cerebellum [77] confirmed binding of these TFs to sites in the vicinity of the *Neurod2* gene (Fig 7A). To address if the Nfi TFs bind the same sites in the developing central nervous system, we asked if these sites are marked by Histone-3-Lysine-27-acetylation (H3K27ac), which labels active enhancers, using e11.5 and e13.5 H3K27ac ChIP-seq data from the forebrain, midbrain, hindbrain, and spinal cord from the ENCODE project [78]. In each region, H3K27ac accumulated around Nfi-bound sites between e11.5 and e13.5 (Fig 7B), thus temporally coinciding with the induction of Nfia and Nfib throughout the nervous system (Figs 2A and 5A and 5B). These data support the hypothesis that the Nfi TFs bind the same sites in the vicinity of the *Neurod2* gene as in the cerebellum and directly regulate its expression in large regions of the developing nervous system.

We thus focused on assaying Neurod2 expression to determine the importance of Nfia and Nfib for the generation of late-born neurons in our in vitro cultures. As our previous characterizations revealed the highest proportion of Neurod2-positive neurons are generated in ventral differentiations (Fig 4B), we focused on this condition. Characterizing the proportion of neurons expressing Neurod2 by flow cytometry and immunofluorescence revealed a marked reduction in the percentage of Neurod2-positive neurons in *Nfia; Nfib* double mutants (Figs 7C, 7D, S14E and S14F). These observations led us to ask if the loss of Nfia and Nfib causes prolonged generation of Zfhx3-positive intermediate neurons. Quantification of Zfhx3-positive neurons by flow cytometry did not reveal an increase in the proportion of Zfhx3-positive neurons (Fig 7C and 7D). Taken together, these data suggest that Nfia and Nfib are required the expression of Neurod2 in late-born neurons but that their activity is not necessary to terminate the phase during which Zfhx3-positive neurons are generated. These data support a model in which the specification of late-born neuronal subtypes is tightly coupled to the signals and transcriptional programs that mediate the switch from neuro- to gliogenesis throughout the nervous system.

## Discussion

### Neuronal diversity from the superposition of spatial and temporal patterning programs

Here, we provide evidence of a global temporal patterning program that operates throughout the vertebrate nervous system to allocate neuronal identity. This functions in parallel to spatial

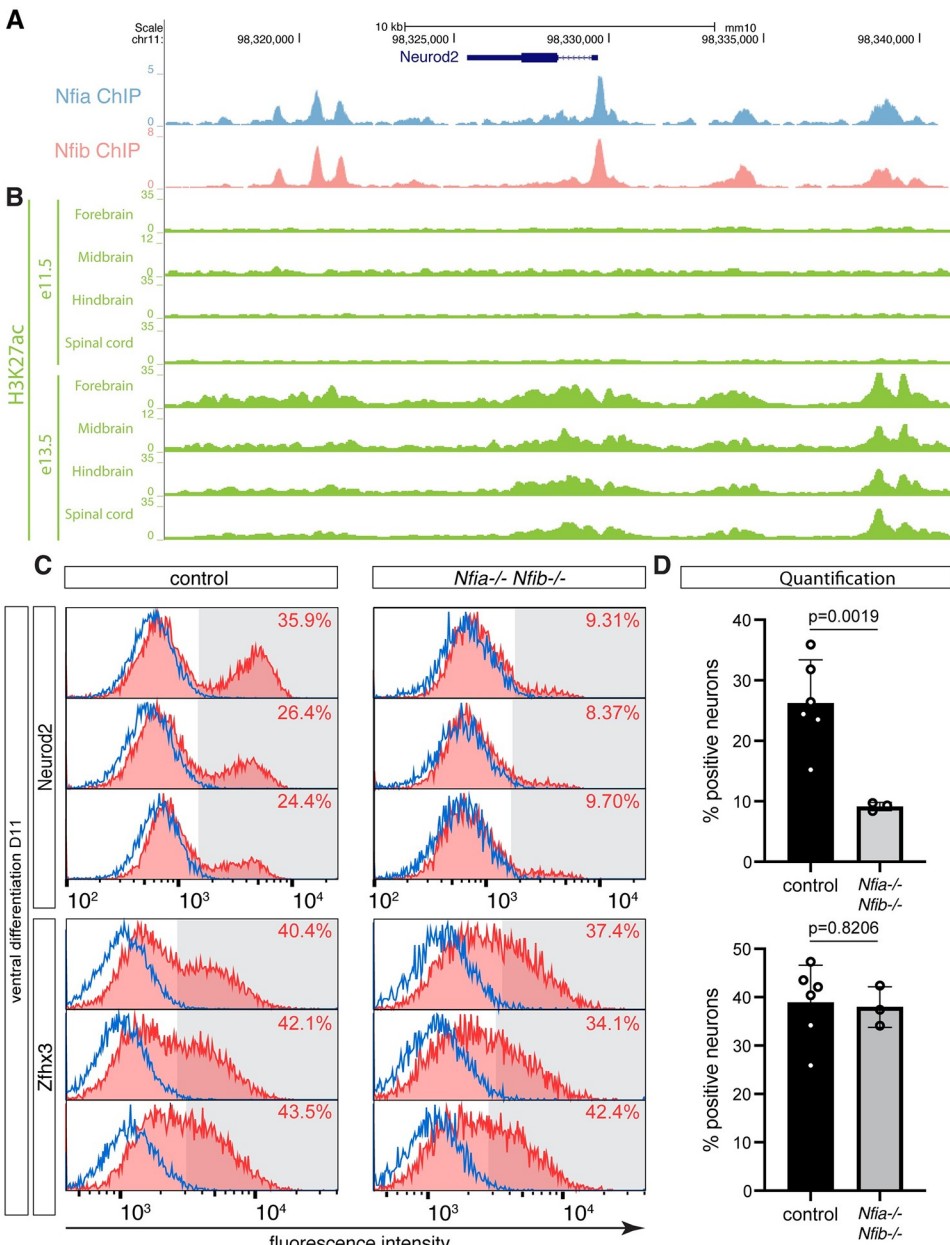

**Fig 7. Nfia and Nfib are required for the efficient generation of late-born Neurod2 neurons (see also S14 Fig).** (A) Analysis of Nfia and Nfib ChIP-Seq data from the mouse cerebellum [77] confirms binding of Nfia and Nfib to genomic regions in the vicinity of the *Neurod2* gene. (B) Analysis of ChIP-seq data from the ENCODE project [78] reveals accumulation of H3K27ac at the same sites between e11.5 and e13.5 in the different regions of the nervous system. (C) Neurod2 (top) and Zfhx3 (bottom) intensity histograms in control (left) and *Nfia*; *Nfib* double mutant (right) neurons (red) and progenitors (blue) at D11 in ventral conditions. Shading indicates the applied thresholds above which cells were counted as Neurod2 or Zfhx3-positive. (D) Percentage of Neurod2 (top) and Zfhx3-positive neurons (bottom) at D11 in control and *Nfia*; *Nfib* double mutants differentiated in ventral conditions (*n* = 6 for control and *n* = 3 for *Nfia*; *Nfib* double mutants). Significance was assessed by unpaired *t*-test with Welch's correction. Underlying flow cytometry data are provided in S3 Data. H3K27ac, Histone-3-Lysine-27-acetylation.

transcriptional programs that pattern the dorsal–ventral and rostral–caudal axes [2,79,80]. The intersection of multiple patterning systems that act along orthogonal axes enables the combinatorial specification and organization of cell types. This suggests how the complexity and diversity of cell types arises from relatively simple patterning schemes [81,82].

The superposition of temporal and spatial patterning programs enables the generation of a combinatorially increasing number of neuronal subsets from a limited number of TFs. Spatial and temporal factors may also directly cross-regulate each other. For example *Nfia* expression in MNs is directly induced by the binding of the somatic MN determinants Lhx3 and Isl1 to the e96 distal *Nfia* enhancer [83], explaining why *Nfia*, generally restricted to late-born neurons, is widely expressed in early-born somatic MNs. Such cross-regulation between spatial and temporal gene regulatory networks provides additional flexibility to tailor temporal TF expression to the specific requirements within a neuronal lineage.

The temporal TF code could further diversify the number of neurons generated in each domain based on the combinatorial coexpression of distinct pairs of temporal TFs. For example, the temporal patterning of *Drosophila* medulla neuroblasts is defined by the sequential expression of 5 TFs; however, the expression of these temporal TFs in aging neuroblasts is not mutually exclusive—instead, there are periods of coexpression of sequentially expressed TFs [15]. Similar observations have been made in the neuroblast lineages in the *Drosophila* embryo and mushroom body [84,85]. Such coexpression of temporal TFs has been proposed to designate additional temporal windows during which further neuronal subtypes are generated [15,84]. Of note, in vertebrates, the respective temporal identities are defined by coexpression of multiple orthologous TFs, which further expands the potential gene regulatory circuitry for the generation of specific neuronal subtypes. Consistent with this idea, characterization of spinal V1 interneuron diversity has revealed differential expression of Onecut1 and Onecut2 in some V1 subtypes, some of which also expressed Zfhx4 [5].

## Temporal factors and the control of neuronal identity

In *Drosophila*, the expression of many temporal TFs is maintained in mature neurons [12]. Similarly, the mutually exclusive expression of Zfhx3 and Nfib/Neurod2 is maintained into the adult nervous system [41]. In other instances, however, the expression of temporal TFs is only transient. The number of neurons expressing Onecut TFs, for example, decreases in many dorsal neuronal populations in the embryonic spinal cord throughout development [37], and the expression of several early TFs, such as Onecut2 and Pou2f2, is not maintained in retinal ganglion cells (S3D Fig). Similarly, many of the TFs in the spatial program are only transiently expressed but have important functions that define the entire subsequent development of a neuronal lineage [86]. Furthermore, at least in some cases, maintaining the expression of temporal TFs in *Drosophila* is not necessary for the maintenance of neuronal identity and function [87]. Genetic tools to permanently label neurons expressing specific temporal TFs during development and to perturb temporal TF function in specific neuronal lineages will be required to carefully map how temporal patterning during development contributes to the generation of neuronal diversity.

Our data provide evidence for the existence of a temporal TF code that applies to large regions of the developing nervous system. These observations contrast with temporal patterning in *Drosophila*, where neuroblasts, and the subsequent neurons they generate, in different parts of the animal, such as embryonic and optic lobe neuroblasts, are patterned by different sets of TFs [11,12]. However, while we focus here on the broad conservation of this temporal program, there is also evidence for additional, tissue-specific temporal patterning mechanisms in mammals. Temporal patterning in the retina, for example, depends on the TFs Ikzf1 and Casz1, which specify early and late identities respectively in this system [88,89]. How the

activities of these TFs are connected to the global temporal patterning program is unclear. Another example is the sequential generation of cortical excitatory neurons. Most cortical excitatory neurons express Nfia/b/x, Neurod2/6, and Tcf4 (S8 Fig), synonymous with a late temporal identity. This is consistent with the late onset of neurogenesis in the cortex compared to other regions of the nervous system and suggests that temporal patterning, at least in this case, does not scale with respect to the neurogenic period. Moreover, cortical neurons are partitioned further into distinct molecular and functional identities by additional temporal transcriptional programs that define cortical layer identity [18,20,61]. Thus, there seems to be an additional temporal patterning program that further partitions late cortical neuronal identity into molecularly and functionally distinct subsets of neurons. Such a mechanism would resemble subtemporal patterning programs identified in *Drosophila* neuroblast lineages [12,90], and it will be interesting to see whether similar subtemporal patterning programs exist in other regions of the vertebrate nervous system.

The temporal TFs may also contribute to the generation of neuronal diversity by subsequently restricting the expression of specific members of early, intermediate, and late TFs to specific neuronal subsets at later developmental stages. Such a mode of action is supported by recent observations in retinal amacrine cells. Late-born glycinergic and non-glycinergic, non-GABAergic amacrine cells express the late temporal TF Tcf4 [55,91], while the expression of other late temporal TFs, such as Neurod2, Neurod6, Nfib, and Nfix, is restricted to specific subsets of the late-born amacrine cells [55]. A similar segregation in expression has been recently demonstrated for Zfhx3 and Zfhx4 in ON and OFF starburst amacrine cells [92].

## A temporal program in progenitors presages the neuronal program

Concomitantly with neurons, neural progenitors throughout the vertebrate nervous system undergo a temporal patterning program (Fig 5A and 5B). Components of this program, including Sox9 and Nfia/b, have previously been implicated in the transition from neurogenesis to gliogenesis [67,68,75,93,94]. However, the expression of these factors precedes the onset of gliogenesis. The expression of Sox9 in neural progenitors coincides with the switch from early Onecut2-positive to intermediate Pou2f2 and Zfhx3-positive neurons, and the induction of Nfia/b correlates with the later transition. Moreover, the loss of generation of late neuronal subtypes in neural progenitors lacking Nfia/b is consistent with the involvement of these TFs in the neuronal temporal program as well as the gliogenic switch (Fig 7). This raises the possibility that the transition of neural progenitors from exclusively neurogenesis to subsequent gliogenesis is part of the same temporal patterning program operating in the nervous system. This would be analogous to the temporal program in *Drosophila* neuroblasts, which also controls the identity of neurons and glia cells.

## TGFβ signaling regulates temporal patterning in the nervous system

Our results indicate that the TGFβ pathway is an important regulator of the pace of progenitor maturation and the timing of temporal TF expression in neurons. This agrees with previous findings. In the hindbrain, TGFβ2 signaling controls the timing of the switch from MNs to serotonergic neurons by repressing the TF Phox2b in neural progenitors [48]. In addition, another TGFβ family member, Gdf11, controls the timing of retinal ganglion cell specification in the vertebrate retina, the timing of MN subtype specification, and the switch from dI5 to late-born dIL$_B$ neurons in the spinal cord [53,95]. The connection between these roles of TGFβ signaling and its role in directing the temporal patterning programs of progenitors and neurons is currently unclear, but, taken together, the results implicate multiple ligands of the TGFβ family in controlling the temporal patterning of the mammalian nervous system. Notably, Activin signaling is involved in controlling the timing of fate switches in the *Drosophila*

mushroom body, and, similar to observations in vertebrates, inhibition of Activin signaling results in a delay of temporal fate progression in this system [49]. These findings suggest a deep evolutionary origin for the role of the TGFβ pathway in controlling temporal patterning and the diversification of cell types in the developing nervous system of bilaterians.

The timing of switches in temporal TF expression take place at approximately similar times throughout the developing nervous system and during the in vitro differentiation of neurons with different axial and dorsal–ventral identities. This raises the question how signals, from locally secreted sources, achieve an apparently globally synchronized effect and what the source of TGFβ might be in the in vitro differentiations. A solution to this puzzle could be that Gdf11 and related ligands are expressed in new-born neurons [53]. Such a model, in which the temporal progression of progenitors is coupled to neurons secreting a ligand that signals back to progenitors, has the advantage that it provides a means to ensure that the correct proportion of neurons with a specific temporal identity are produced before progenitors switch to the next phase. A prediction of such a model is that local increases in neurogenesis would lead to a local acceleration of temporal patterning in progenitors. Indeed, several genes involved in the onset of gliogenesis, such as *Sox9*, *Nfia*, and *Fgfr3*, are first expressed in the ventral spinal cord [67,68,75], where MNs differentiate at higher rate at early developmental stages [96,97]. Further experiments that explore the connection between Gdf11, neurogenesis rate, and temporal patterning are required to test this hypothesis.

The data show that the temporal pattern of both neurons and progenitors continues to advance in the absence of TGFβ pathway activity. This is consistent with observations from the ventral hindbrain, where ablation of Tgfbr1 delays but does not abrogate the switch from MNs to serotonergic neurons [48,98], and in Gdf11 mutants in the spinal cord, where the onset of oligodendrocyte formation is delayed but not prevented [53]. Together, this suggests that other extrinsic signals, or cell-intrinsic timers, must exist that promote temporal progression. A potential candidate signal that may oppose the activity of TGFβ is retinoic acid (RA), which has been shown to drive the generation of Onecut-positive Renshaw cells in an in vitro model of V1 subtype diversity [99]. Furthermore, the rate-limiting enzyme for RA synthesis is down-regulated in somites, adjacent to the neural tube, between e9.5 and e10.5 [100], coinciding with the switch from Onecut to Zfhx3-positive neurons. In addition, several pathways, such as Neuregulins, Notch, FGF, and JAK/STAT, have been shown to promote gliogenesis [23,94,101]. Given the pivotal role of Nfi TFs in this process, one or more of these signals may promote the acquisition of a late Nfi-positive progenitor identity. The genetic and experimental accessibility of in vitro models will allow these possibilities to be tested.

## Temporal patterning of in vitro generated neurons

In vitro generated neurons are widely used for disease modeling and have the potential to offer novel therapeutic avenues to tackle nervous system injuries and neurodegenerative diseases [102–104]. A better understanding of the molecular mechanisms responsible for neuronal diversity contributes to the rational design of in vitro differentiation protocols to generate cell types best suited for such applications. Our work demonstrates that the temporal patterning of neurons and progenitors is conserved in vitro, providing a new dimension for assessing the identity of progenitors and neurons obtained in culture. Furthermore, the observation that manipulating TGFβ signaling can accelerate or slow down the progression of temporal patterning opens up the possibility to use such perturbations to increase the yield of progenitors and neurons with desired spatial and temporal identities.

Many applications of in vitro generated neurons and progenitors require large numbers of cells with defined identities. These are often generated by expanding progenitors using

treatments with signals such as EGF and/or FGF before exposing the resulting progenitor populations to differentiation stimuli. Such prolonged expansion phases might result in the preferential generation of neurons with late temporal identities. This might be at least partially counteracted by the incorporation of TGFβ pathway inhibitors. Indeed, treatment with SB431542 in combination with other small molecules has been demonstrated to enable long-term self-renewal of neural stem cells [105]. Another promising approach to generate neurons with defined identities is reprogramming of pluripotent cells or somatic cells, such as fibroblasts or astrocytes, using specific cell fate–converting cocktails of transcription regulators. Notably, the reprogramming of ES cells to different types of neurons results in expression of Onecut TFs [106–109], suggesting that such approaches might preferentially generate the earliest temporal identities. The addition of temporal TFs that define later stages of the differentiation program to these reprogramming cocktails might expand the toolbox for the efficient generation of a wider range of neuronal subtypes with desired temporal identities for in vitro disease modeling and future clinical applications.

## Experimental procedures

### Animal welfare

Animal experiments were performed under UK Home Office project licenses (PD415DD17) within the conditions of the Animal (Scientific Procedures) Act 1986. All experiments were conducted using outbred UKCrl:CD1 (ICR) (Charles River) mice.

### Immunofluorescent staining and microscopy

Embryos were fixed at the indicated stages in 4% PFA (Thermo Fisher Scientific) in PBS on ice, cryoprotected and dissected in 15% ice-cold sucrose in 0.12 M PB buffer, embedded in gelatine and 14 μm sections taken. In vitro generated cells were fixed for 15 minutes in 4% PFA in PBS at 4 degrees. Approximately 30 minutes blocking and primary antibody incubation over night at 4 degrees was performed using PBS + 0.1% Triton (PBS-T) + 1% BSA. A complete list of antibodies is available in S1 Table. The next day, samples were washed 3× 30 minutes in PBS-T and incubated with secondary antibodies in PBS-T + 1% BSA for 1 hour at room temperature. Secondary antibodies used throughout the study were raised in donkey (Life Technologies, Jackson Immunoresearch). Alexa488- and Alexa568-conjugated secondary antibodies were used at 1:1,000, Alexa647-conjugated antibodies at 1:500. Samples were washed 3 more times in PBS-T and then mounted in Prolong Antifade (Molecular Probes).

For EdU labeling, mice were intraperitonially injected with 3 μl/gramm body weight EdU diluted in PBS at the indicated stages. EdU was detected using Alexa647 Click-iT EdU Imaging Kit (Invitrogen C10340) according to the manufacturer's specifications. At least 3 sections from different animals were analyzed for each time point.

Stainings of in vitro differentiations were acquired on a Zeiss Imager.Z2 microscope equipped with an Apotome.2 structured illumination module and a 20× air objective (NA = 0.75). Cryosections were imaged using a Leica SP8 equipped with a 40× oil PL APO CS2 objective (NA = 1.30) or a Zeiss LSM880 equipped with a Fluar 40× oil M27 objective (NA = 1.30). Tissue sections were tiled using 5% or 10% overlap between adjacent tiles and merged using LAS X or ZEN Black software.

### Image analysis

Image analysis was performed in Fiji (http://fiji.sc/Fiji) and Python3.7 (http://www.python.org). e13.5 mouse neural tube transverse sections were manually cropped using Fiji and then

processed using a custom Python pipeline. Cell nuclei were segmented using an adaptive threshold and watershed algorithm on the DAPI channel. Parameters for proper segmentation and filtering were manually optimized for each set of images. Segmented objects were further filtered based on area to fit the expected nuclei dimensions. Neuronal nuclei were distinguished from those of progenitors either by presence of the neuronal marker HuC or absence of Sox2 staining. For each neuronal nucleus, the mean intensity of the temporal TFs and EdU was then calculated (S1 and S2 Data).

Data analysis and plotting were performed in R (https://www.r-project.org). For each section, intensities in nuclei were first normalized between 0 and 1. To remove outliers, 0.3% of the brightest and dimmest objects were discarded. Objects were counted as positive for EdU or expression of temporal TFs if their normalized intensity was greater than 0.25. Percentage of EdU-positive nuclei expressing temporal TFs was then plotted using ggplot2 [110] (S1 and S2 Data).

## ES cell culture and differentiation

HM1 mouse ES cells (Thermo Fisher Scientific) were maintained and differentiated as described previously [64–66]. In brief, ES cells were maintained on a layer of mitotically inactivated mouse embryonic fibroblast (feeders) in ES cell medium + 1,000 U/ml LIF. For differentiation, ES cells were dissociated using 0.05% Trypsin (Gibco). Feeder cells were removed by replating cells for 25 minutes on a tissue culture plate. About 60 to 80,000 cells were plated onto 0.1% Gelatin (Sigma) coated 35 mm CellBIND dishes (Corning) into N2B27 medium + 10 ng/ml bFGF. Differentiation protocols for progenitors and neurons with different axial and dorsal–ventral identities are shown in Fig 4A. Differentiation protocols for activation and inhibition of the TGFß pathway using TGFß2 (R&D Systems) or 10 μM SB431542 (Tocris), respectively, are shown in Fig 6A and 6F. For midbrain differentiation, cells were kept in N2B27 medium with addition of 10 ng/ml bFGF until day 3. To generate hindbrain identity, 100 nM RA (Sigma) and 500 nM SAG (Calbiochem) were supplemented together at days 3 and 4. For spinal cord differentiations, cells were exposed to 5 μM CHIR99021 (Axon) between days 2 and 3 and then supplemented with 100 nM RA (Sigma) until day 5. For ventral differentiations, cells were additionally exposed to 500 nM SAG (Calbiochem) from days 3 to 9.

## Generation of *Nfia; Nfib* double-mutant ESCs

For generation of *Nfia; Nfib* double-mutant ES cells, CRISPR guide RNAs were cloned into pX459 plasmid obtained from Addgene (# 62988), according to [111]. ES cells were electroporated using Nucleofector II (Amaxa) and mouse ESC Nucleofector kit (Lonza). Afterwards, cells were replated onto 10-cm CellBind plates (Corning) and maintained in 2i medium + LIF. For selection, cells were first treated with 1.5 μg/ml Puromycin (Sigma) for 2 days and afterwards maintained in 2i medium + LIF until colonies were clearly visible. Individual colonies were picked using a 2-μl pipette, dissociated in 0.25% Trypsin (Gibco), and replated onto feeder cells in ES medium + 1,000 U/ml LIF in a 96-well plate. Mutations in *Nfia* and *Nfib* were analyzed by PCR over the targeted regions and verified by Sanger sequencing. Overlapping peaks arising from heterozygous indels were deconvolved using CRISP-ID [112] (S14A and S14B Fig). Loss of Nfia and Nfib protein was further confirmed by immunofluorescent staining at day 10 of the differentiation (S14C and S14D Fig).

## Flow cytometry

In vitro differentiations were dissociated at the indicated time points using 0.05% Trypsin (Gibco). Live/Dead cell staining was performed using LIVE/DEAD Fixable Near-IR Dead Cell

Stain Kit (Invitrogen) for 30 minutes on ice. Immediately, afterwards, cells were spun down for 2 minutes at 1,000×*g* and fixed in 4% PFA for 12 minutes on ice. Fixed cells were spun down, resuspended in 500 µl PBS, and stored at 4 degrees for up to 2 weeks.

For staining, 1.5 to 2 million cells were used. Cells were spun down and incubated with antibodies in PBS-T + 1% BSA. If primary and secondary antibodies were used, cells were incubated in primary antibody solution over night. Directly conjugated antibodies or secondary antibodies were applied for 1 hour at room temperature. A complete list of antibodies used for flow cytometry is supplied in S2 Table. Flow cytometry analysis was performed using BD LSR Fortessa analyzers (BD Biosciences). Data analysis was performed using FlowJo (v10.4.1) and plotted using Graphpad Prism 7. The general gating strategy is outlined in S11B Fig. Progenitor and neuronal cell populations were discriminated based on Sox2 and Tubb3 antibody staining (S11B Fig). Percentages of Onecut2, Neurod2, and Zfhx3-positive neurons were calculated by applying a threshold at which 1% to 2% of Sox2-positive progenitors in the same sample were counted as positive. Percentage of Nfia-positive neurons and progenitors was determined using a global threshold for all datasets. Data were plotted and statistical analysis performed in GraphPad Prism 8. Graphs throughout the manuscript show means ± standard deviation of all conducted replicates. Statistical significance was assessed using unpaired *t* tests. A summary of the percentage of positive cells, replicate number, and *p*-values is provided in S3 Data. Significance values throughout the manuscript are indicated by $p < 0.001 = ^{***}$, $p < 0.01 = ^{**}$, $p < 0.05 = ^{*}$.

## RT-qPCR

Total RNA was isolated from cells at the indicated time points using Qiagen RNeasy kit according to the manufacturer's instructions. Genomic DNA was removed by digestion with DNase I (Qiagen). cDNA synthesis was performed using SuperScript III (Invitrogen) and random hexamers. qPCR was performed using PowerUp SYBR Green Master Mix (Thermo Fisher Scientific) using 7900HT Fast Real time PCR (Applied Biosystems), QuantStudio 5 or QuantStudio 12K Flex Real-Time PCR Systems (Thermo Fisher Scientific). qPCR primers were designed using NCBI tool Primer BLAST and are listed in S3 Table. All experiments were conducted at least in biological triplicates for each time point analyzed. Expression values were normalized to ß-actin. Data were plotted and statistical analysis performed in GraphPad Prism 8. Graphs throughout the manuscript show means ± standard deviation of all replicates. A summary of all qPCR data is provided in S4 Data.

## scRNAseq data analysis

scRNAseq analysis was performed in R-Studio using R v3.5.2 and later. Scripts describing the scRNAseq analysis performed in this paper are available at https://github.com/andreassagner/tTF_paper_2020.

**Differential gene expression analysis in spinal cord neural progenitors.**   scRNAseq data from e9.5 to e13.5 spinal cord neural progenitors including subtype annotations were obtained from Delile and colleagues. dp6 progenitors were excluded from this analysis due to low numbers in the dataset. For each progenitor domain, differential gene expression between progenitors from different embryronic days was performed using Seurat [113] using the "FindAllMarkers" function with settings min.pct = 0.25 and logfc.threshold = 0.25. Only genes detected in more than 7 progenitor domains were retained. TFs were identified based on a list of TFs encoded in the mouse genome obtained from AnimalTFDB3.0 [114]. Heatmaps in Fig 5A show log-scaled and z-scored gene expression.

**Analysis of temporal TFs in the mouse retina.** scRNAseq of the developing retina [43] was downloaded from https://github.com/gofflab/developing_mouse_retina_scRNASeq and imported into Seurat v3.1.4 [113]. Cells were filtered based on age (e14, e16, e18, and P0), cell type (RPCs, Neurogenic Cells, Photoreceptor Precursors, Cones, Rods, Retinal Ganglion Cells, Amacrine Cells, Horizontal Cells), number of reads in each cell (nFeature > 800 and nFeature < 6,000), and percentage of reads in mitochondrial genes (percent.mt < 6). Only cells annotated as Horizontal Cells, Amacrine Cells, Retinal Ganglion Cells, Rods, and Cones were used for the time-stratified heatmap of temporal TF expression.

**Expression dynamics of temporal TFs in the forebrain, midbrain, and hindbrain.** Annotated scRNAseq data from the developing forebrain, midbrain, and hindbrain were downloaded from mousebrain.org [9]. Cells were assigned forebrain, midbrain, and hindbrain identity based on the "Tissue" column of the provided loom file. To account for the different sequencing depths between cells, readcounts were normalized by multiplying the counts in each cell with 10,000 divided by the total number of UMIs in this cell. Mean expression and ratio of expressing cells for the indicated temporal TFs and regions were calculated in R. Data were plotted using ggplot2 [110]. Heatmaps in Fig 5B show log-scaled and z-scored gene expression.

**Pseudotemporal ordering of gene expression dynamics.** A detailed description how the pseudotemporal ordering was performed is provided as S1 Text. Cell annotations from Delile and colleagues were used for spinal cord scRNAseq data. For identification of neuronal lineages in the hindbrain and forebrain, scRNAseq data from La Manno and colleagues were analyzed by UMAP dimensionality reduction. Cell neighborhoods and clustering were determined using the standard Seurat workflow using the FindNeighbors and FindClusters functions [113]. Pseudotemporal ordering on neuronal lineages was performed using the Slingshot R package [115] using UMAP dimensionality reduction performed in Seurat. Start clusters for pseudotime reconstruction were chosen based on high expression of proneural bHLH TFs. Clusters most distant on the UMAP from the respective start clusters were chosen as end clusters for the Slingshot algorithm. In a few cases, curves predicted by the Slingshot algorithm for pseudotime reconstruction were excluded, e.g., if they curved back on themselves or crossed other Slingshot curves (see S1 Text). For each developmental stage, gene expression along pseudotime, defined by the remaining pseudotime curves, was fitted using LOESS regression implemented in the ggplot2 R package [110]. Pseudotime was then subdivided into 30 pseudotime bins, and gene expression along the trajectory was normalized for each gene across all time points.

**Comparison with in vitro RNAseq data.** RNAseq data from D3 to D10 ventral spinal cord differentiations [69] (GSE140748) were used. Gene expression per time point was averaged over all 3 provided replicates. Only data from full days of differentiation (D3, D4, D5, D6, D7, D8, D9, D10; D0 to D7 in the provided data files) were used for further analysis. Heatmaps in Fig 5C show log-scaled and z-scored gene expression. To identify correlated, uncorrelated, and anticorrelated genes in S13A–S13C Fig, log-scaled, z-scored gene expression data of the 542 genes identified in Fig 5 were compared between e9.5 and e13.5 in vivo neural progenitors [33] and days 5 to 9 from the in vitro differentiations (D2 to D6 in the provided data files) by Pearson correlation (S5 Data).

## Nfia/b/x and H3K27ac ChIP-seq data

Nfia/b/x ChIP-seq data from Fraser and colleagues were downloaded from the GEO database (GSE146793) and aligned to mm10 using the nf-core ChIP-seq pipeline v1.1.0 [116]. H3K27ac ChIP-seq tracks were obtained via the UCSC genome browser and ENCODE DNA trackhub. H3K27ac tracks show signal fold change over control.

## Supporting information

**S1 Fig. Related to Fig 1: Complete time course of colocalization between temporal TFs and EdU administered at different time points.** (A-C). Colocalization between Zfhx3 (A), Nfib (B), Neurod2 (C), and EdU administered at e9.5, e10.5, e11.5, or e12.5 (from left to right) in e13.5 spinal cord sections. Scale bars in overview pictures = 200 μm, insets = 50 μm. TF, transcription factor.
(PNG)

**S2 Fig. Related to Fig 1: Nonoverlapping expression of temporal TFs in spinal cord neurons at e13.5.** (A, B) Zfhx3 and Nfib (A) or Zfhx3 and Neurod2 (B) are expressed in mutually exclusive populations of neurons in the spinal cord. Scale bars in overview pictures = 200 μm, insets = 20 μm. TF, transcription factor.
(PNG)

**S3 Fig. Related to Fig 2: Characterization of temporal TF expression in the developing retina.** (A) UMAP representation of scRNAseq data from the developing mouse retina [43] color coded by developmental stage. (B) Same UMAP representation as (A) color coded for cell identity. (C) Expression levels of *Onecut2*, *Pou2f2*, *Zfhx3*, and *Nfib* in individual cells. (D) Heatmap indicating expression levels of the temporal TFs (*Onecut2*, *Pou2f2*, *Zfhx3*, and *Nfib*) and known marker genes (*Lhx1*, *Pax6*, *Pou4f2*, *Thrb*, and *Nrl*) in different types of retinal neurons stratified by developmental age. AC, amacrine cell; HC, horizontal cell; RGC, retinal ganglion cell; RPC, retinal progenitor cell; NCs, neurogenic cell; PP, photoreceptor precursor; scRNAseq, single-cell RNA sequencing; TF, transcription factor; UMAP, Uniform Manifold Approximation and Projection.
(PNG)

**S4 Fig. Related to Fig 2: Characterization of temporal TF expression in the developing central nervous system.** (A-D). UMAP representation of scRNAseq data from (A) forebrain, (B) midbrain, (C) hindbrain [9], and (D) spinal cord [33] color coded by developmental stage. (E-H) Expression levels of *Slc17a6*, *Gad2*, *Onecut2*, *Pou2f2*, *Zfhx3*, *Nfia*, *Nfib*, and *Neurod2* in individual cells. (I-L) Heatmaps indicating Spearman correlation between temporal TF expression in the different regions of the nervous system. (M) Spearman correlation rank plots for *Nfib* (top row) and *Zfhx3* (bottom row) in the scRNAseq data from forebrain, midbrain, hindbrain, and spinal cord (left to right). Data points corresponding to temporal TFs are highlighted in red. scRNAseq, single-cell RNA sequencing; TF, transcription factor; UMAP, Uniform Manifold Approximation and Projection.
(PNG)

**S5 Fig. Related to Fig 2: Nfib-positive cells in the hindbrain mantle layer are neurons.** (A-C) e13.5 hindbrain sections stained for Nfib and the progenitor marker Sox2 (A), the glial progenitor marker Sox9 (B), and the neuronal marker Lhx5 (C). Scale bars in overview pictures = 200 μm, insets = 25 μm.
(PNG)

**S6 Fig. Related to Fig 2: Pseudotemporal ordering confirms sequential generation of temporal TF expressing neurons.** (A) Sequential generation of neurons expressing temporal TFs should result in the capture of neurons at different stages of their differentiation trajectory in scRNAseq time course data. (B) Pseudotime reconstruction of gene expression dynamics along a neuronal differentiation trajectory. Dark blue corresponds to early cells, yellow to late cells along the differentiation trajectory. Arrows indicate predicted pseudotime trajectories. (C) Temporal TFs should be sequentially expressed in pseudotime. (D) Pseudotime

reconstruction of gene expression for different neuronal lineages along the dorsal–ventral axis of the spinal cord reveals sequential expression of temporal TFs. (E, F) Similar gene expression dynamics are observed when pseudotemporal gene expression is reconstructed for neuronal lineages in the hindbrain (E) and forebrain (F). LGE, lateral ganglionic eminence; MGE, medial ganglionic eminence; scRNAseq, single-cell RNA sequencing; TF, transcription factor.
(PNG)

**S7 Fig. Related to Fig 2: EdU birthdating confirms sequential generation of Zfhx3 and Nfib-positive neurons in the midbrain and hindbrain.** (A-D) e13.5 hindbrain (A, B) and midbrain (C, D) sections stained for Zfhx3 (green), EdU (red), and Sox2 (blue). EdU was administered at e10.5 (A, C) or e12.5 (B, D). (E-H) e13.5 hindbrain (E, F) and midbrain (G, H) sections stained for Nfib (green), EdU (red), and Sox2 (blue). EdU was administered at e10.5 (E, G) or e12.5 (F, H). Scale bars in overview pictures = 100 μm, insets = 25 μm.
(PNG)

**S8 Fig. Related to Fig 2: Widespread expression of late temporal TFs in cortical glutamatergic neurons.** (A) UMAP plots of all e10–e13.5 forebrain neurons in the dataset from La Manno and colleagues. Cortical excitatory neurons are colored in red. (B) UMAP plots indicating the expression of marker genes characteristic for cortical excitatory neurons. (C) UMAP plots showing widespread expression of late temporal TFs in forebrain excitatory neurons (top row) and expression of marker genes for cluster 7 neurons (see D) (bottom row). (D) Identification of different clusters of cortical excitatory neurons. Cluster 7 corresponds to the Zfhx3-positive population of neurons (see also C). (E) Differential gene expression analysis comparing cluster 7 cells to the rest of the identified cortical excitatory neurons. The top 6 TFs up-regulated in this cluster are indicated by the red box, the top 6 down-regulated TFs by the blue box. (F) UMAP plot of cortical excitatory neurons (red cells in A) color coded for the developmental stage from which these cells were obtained. TF, transcription factor; UMAP, Uniform Manifold Approximation and Projection.
(PNG)

**S9 Fig. Related to Fig 2: Differential expression of intermediate and late temporal TFs in scRNAseq data from the late forebrain and midbrain.** (A, F) UMAP plots from late midbrain (A) and forebrain (F) neurons (e16–e18) color coded for the developmental stage from which these cells were obtained. (B, G) UMAP plots from late midbrain (B) and forebrain (G) neurons showing expression of the indicated markers. Expression of intermediate and late temporal markers (especially *Zfhx3/4* and *Nfia/b*) stay highly anticorrelated in neurons. (C, H) *Nfia* and *Nfib*-positive cells express the neuronal markers *Tubb3* and *Elavl3* but not the glial markers *S100b* and *Slc1a3*. (D, I) Heatmaps indicating Spearman correlation between intermediate and late temporal TFs in the late embryonic midbrain (D) and forebrain (I). (E, J) Correlation rank plots for *Zfhx3* and *Nfib* indicate that these markers stay highly anticorrelated in the late embryonic midbrain (E) and forebrain (J). scRNAseq, single-cell RNA sequencing; TF, transcription factor; UMAP, Uniform Manifold Approximation and Projection.
(PNG)

**S10 Fig. Related to Fig 3: Midbrain red nucleus neurons express Zfhx3 but not Onecut2.** (A, B) e11.5 midbrain cryosection stained for (A) Onecut2 (red) and Pou4f1 (green) or (B) Zfhx3 (red) and Pou4f1 (green). Note the Onecut2 expression is mutually exclusive with Pou4f1 expression (A) while most Pou4f1 neurons express Zfhx3 (B). Scale bars in overview pictures = 200 μm, insets = 25 μm.
(PNG)

**S11 Fig. Related to Fig 4: Further characterization of the in vitro differentiations.** (A) RT-qPCR analysis of *Foxg1*, *Otx2*, *Hoxa4, Hoxb9*, and *Hoxc8* reveals the generation of neurons and progenitors with different axial identities in the in vitro differentiations. For underlying data, see S4 Data. (B) Gating strategy for the quantification of the expression of different markers in neurons and progenitors by flow cytometry. Living cells were identified based on Infrared Life/Dead stain. Gating on single cells was achieved using forward and side scatter as indicated. Progenitors and neurons were discriminated based on the progenitor marker Sox2 and neuronal beta-tubulin (Tubb3). To quantify the proportion of neurons expressing Onecut2, Zfhx3, and Neurod2, an intensity threshold was applied to each sample that was exceeded by 1%–2% of progenitors. The same threshold was then applied to neurons in the same sample, and the percentage of neurons exceeding this threshold was counted as positive. As Nfia is expressed in neurons and progenitors, a global threshold was applied to quantify the proportion of neurons and progenitors expressing Nfia. (C) Characterization of dorsal and ventral spinal cord differentiations by immunostaining. Under dorsal conditions, most neurons express the TF Lbx1, which is expressed in dI4-dI6 neurons generated in the intermediate dorsal part of the spinal cord. Under ventral conditions, neurons express the V3 interneuron marker Sim1. Scale bars in C = 25 μm. MN, motor neuron; RT-qPCR, real-time quantitative polymerase chain reaction; SAG, Shh pathway agonist; TF, transcription factor.
(PNG)

**S12 Fig. Related to Fig 5: Temporal patterning of neurons and progenitors.** (A) Spatial and temporal expression of the 33 differentially expressed TFs during the neurogenic period in spinal cord neural progenitors. DV, dorsal–ventral; TF, transcription factor.
(PNG)

**S13 Fig. Related to Fig 5: Comparison of gene expression dynamics between in vivo and in vitro.** (A-C) Expression dynamics of correlated (A), uncorrelated (B), and anticorrelated (C) genes (left) and TFs (right) in embryonic progenitors (left plots) and RNAseq data from the in vitro differentiations (right plots). Heatmap shows log-scaled and z-scored gene expression values. Pearson correlation values are provided in S5 Data. (D) RT-qPCR analysis for *Lin28b*, *Sox9*, *Npas3*, *Zbtb20*, *Nfib*, and *Hopx* from days 5–11 in in vitro generated differentiations with different axial identities reveals that temporal patterning is conserved in vitro. Underlying data are included in S4 Data. RT-qPCR, real-time quantitative polymerase chain reaction; TF, transcription factor.
(PNG)

**S14 Fig. Related to Fig 7: Characterization of the *Nfia; Nfib* double-mutant ES cell line.** (A, B) Engineering of a *Nfia; Nfib* double-mutant ES cell line by CRISPR/Cas9-mediated mutagenesis. Introduction of double heterozygous frameshift mutations in both genes was validated by Sanger sequencing. (C, D) Loss of Nfia (C) and Nfib (D) immunostaining in neural progenitors generated from *Nfia; Nfib* double-mutant ES cells in dorsal differentiations at D10. (E, F) Reduced number of Neurod2-positive neurons in ventral differentiations of *Nfia; Nfib* double mutants compared to controls at D11 (E) and D13 (F) revealed by immunostaining. Scale bars in C-F = 20 μm. ES, embryonic stem.
(PNG)

**S1 Table. Related to Experimental procedures: Antibodies for immunofluorescence.**
(XLSX)

**S2 Table. Related to Experimental procedures: List of antibodies for flow cytometry.**
(XLSX)

**S3 Table. Related to Experimental procedures: List of primers for RT-qPCR analysis.**
(XLSX)

**S1 Data. Related to Fig 1H: Percentage of EdU-positive neurons expressing the respective temporal TFs in the spinal cord.** Data file also includes intensity measurements of the individual channels from all replicates. TF, transcription factor.
(XLSX)

**S2 Data. Related to Fig 2I–2K: Percentage of EdU-positive neurons expressing the respective temporal TFs in the hindbrain, midbrain, and ventral midbrain.** Data file also includes the intensity measurements of the individual channels from all replicates. TF, transcription factor.
(XLSX)

**S3 Data. Related to Figs 4, 5, 6, and 7: Summary of flow cytometry results.**
(XLSX)

**S4 Data. Related to Figs 5D, 6B, 6G, 6H, S11A, and S13D: Summary of qPCR data.**
(XLSX)

**S5 Data. Related to S13A–S13C Fig: Comparison of gene expression dynamics between embryonic neural progenitors and the in vitro differentiations.**
(XLSX)

**S1 Text. Related to S6 Fig: Reconstruction of pseudotemporal gene expression dynamics for different neuronal populations in the spinal cord, hindbrain, and forebrain.**
(PDF)

## Acknowledgments

We thank all members of the Briscoe lab for help, advice, reagents, and critical feedback. We acknowledge scientific support by the Crick Science and Technology platforms, in particular the Biological Research Facility, Equipment Park, Flow Cytometry, and Light Microscopy facilities. We thank M.J. Delás for help with flow cytometry; Thomas Müller, Carmen Birchmeier, and Siew-Lan Ang for kindly sharing antibodies; Cerys Manning for help with microscopy; and Nancy Papalopulu, Tiago Rito, and François Guillemot for comments on the manuscript.

## Author Contributions

**Conceptualization:** Andreas Sagner, James Briscoe.

**Formal analysis:** Andreas Sagner.

**Funding acquisition:** James Briscoe.

**Investigation:** Andreas Sagner, Isabel Zhang, Thomas Watson, Manuela Melchionda, James Briscoe.

**Methodology:** Andreas Sagner, Isabel Zhang, Thomas Watson, Manuela Melchionda.

**Project administration:** James Briscoe.

**Resources:** Thomas Watson.

**Software:** Jorge Lazaro.

**Supervision:** Andreas Sagner, James Briscoe.

**Validation:** Andreas Sagner.

**Writing – original draft:** Andreas Sagner, James Briscoe.

**Writing – review & editing:** Andreas Sagner, Isabel Zhang, Thomas Watson, Jorge Lazaro, James Briscoe.

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
