## [Editor Report · Decision Letter 0]

10 Jun 2021

Dear Dr Briscoe, 

Thank you for submitting your manuscript entitled "Temporal patterning of the central nervous system by a shared transcription factor code" for consideration as a Research Article by PLOS Biology.

Your manuscript has now been evaluated by the PLOS Biology editorial staff, as well as by an academic editor with relevant expertise, and I am writing to let you know that we would like to give you the opportunity to address the concerns raised by the referees at Review Commons and then re-review. 

However, before we can stamp a major revision decision and give you detailed feedback on the reviewer comments, we need you to complete your submission by providing the metadata that is required for full assessment. To this end, please login to Editorial Manager where you will find the paper in the 'Submissions Needing Revisions' folder on your homepage. Please click 'Revise Submission' from the Action Links and complete all additional questions in the submission questionnaire.

Please re-submit your manuscript within two working days, i.e. by Jun 14 2021 11:59PM.

Kind regards,

Gabriel Gasque

Senior Editor

PLOS Biology

ggasque@plos.org

---

## [Editor Report · Decision Letter 1]

15 Jun 2021

Dear Dr Briscoe,

Thank you very much for submitting your manuscript "Temporal patterning of the central nervous system by a shared transcription factor code" for consideration as a Research Article at PLOS Biology. As mentioned previously, your manuscript has been evaluated by the PLOS Biology editors and by an Academic Editor with relevant expertise. We also considered the reviews you transferred from Review Commons. 

In light of the those reviews, we will not be able to accept the current version of the manuscript, but we would welcome re-submission of a much-revised version that takes into account the reviewers' comments. We cannot make any decision about publication until we have seen the revised manuscript and your response to the reviewers' comments. Your revised manuscript is also likely to be sent for further evaluation by the reviewers.

We expect to receive your revised manuscript within 3 months. 

**IMPORTANT - SUBMITTING YOUR REVISION**

Your revisions should address the specific points made by each reviewer. In addition to those comments, the Academic Editor provided feedback, which should also be addressed and included in your point-by-point response. You can find them below my signature. 

In your cover letter, you mentioned the possibility of splitting your manuscript in two. At this point, and without seeing your final revision and response to reviewers, we cannot say whether that would work for us. Our recommendation is that you submit a single revised paper, and once we read it, we decide how to proceed.

Please submit the following files along with your revised manuscript:

*Re-submission Checklist*

*Published Peer Review*

*PLOS Data Policy*

*Blot and Gel Data Policy*

Sincerely,

Gabriel Gasque

Senior Editor

PLOS Biology

ggasque@plos.org

Comments from the Academic Editor:

- In both neuronal and neuronal progenitor cross-tissue analyses, the authors look for the expression of the temporal TFs from the spinal cord in the datasets of the forebrain, midbrain, and hindbrain. It would be interesting to also look the other way round, i.e. to look for TFs that are differentially expressed in neuronal progenitors of different ages in the brain and see if there are any differences between the brain and the spinal cord.

- In their in vitro neuronal differentiation experiments, the authors rightfully focus on the similarities. It would be very interesting and a useful contribution to the in vitro differentiation field to also highlight the differences, which exist for sure (they only mention the higher proportion of late born neurons, but I would expect more differences to exist).

- The most important thing in my mind would be to improve the quality of the data for the cortex by exploiting very extensive scRNAseq data from the Jabaudon or Arlotta labs and confirm that this is supporting their model.

---

## [Editor Report · Decision Letter 2]

5 Oct 2021

Dear Dr Briscoe,

Thank you for submitting your revised Research Article entitled "Temporal patterning of the central nervous system by a shared transcription factor code" for publication in PLOS Biology. I have now discussed your new version with other staff editors and with the Academic Editor. 

I am pleased to tell you that we will probably accept this manuscript for publication, provided you satisfactorily address the following data and other policy-related requests:

1) Title: We would like to suggest an active title, which we think will be more appealing to a broad readership: “The temporal patterning of the central nervous system is orchestrated by a shared transcription factor code.”

2) Data: You may be aware of the PLOS Data Policy, which requires that all data be made available without restriction: http://journals.plos.org/plosbiology/s/data-availability. For more information, please also see this editorial: http://dx.doi.org/10.1371/journal.pbio.1001797

Note that we do not require all raw data. Rather, we ask for all individual quantitative observations that underlie the data summarized in the figures and results of your paper. For an example see here: http://www.plosbiology.org/article/info%3Adoi%2F10.1371%2Fjournal.pbio.1001908#s5

These data can be made available in one of the following forms:

Regardless of the method selected, please ensure that you provide the individual numerical values that underlie the summary data displayed in the following figure panels: Figures 1G, 2AIJK, 4BCD (Table S3?), 5A-E (Table S3?), 6BC(Table S3?)D(Table S3?)EGH, 7CD(Table S3?), S3A-D, S4A, S5A-M, S6DEF, S8ABCDF, S9ABCDFGHI, S11AB, S12, and S13ABC(Table S5?)D.

2.a) Please also ensure that each figure legend in your manuscript includes information on where the underlying data can be found and that your supplemental data file/s has/have a legend.

2.b) Please ensure that your Data Statement in the submission system accurately describes where your data can be found.

We expect to receive your revised manuscript within two weeks. 

*Published Peer Review History*

*Early Version*

Sincerely,

Gabriel Gasque, Ph.D.,

Senior Editor,

ggasque@plos.org,

PLOS Biology

---

## [Editor Report · Decision Letter 3]

20 Oct 2021

Dear Dr Briscoe,

On behalf of my colleagues and the Academic Editor, Claude Desplan, I am pleased to say that we can in principle offer to publish your Research Article "A shared transcriptional code orchestrates temporal patterning of the central nervous system" in PLOS Biology, provided you address any remaining formatting and reporting issues. These will be detailed in an email that will follow this letter and that you will usually receive within 2-3 business days, during which time no action is required from you. Please note that we will not be able to formally accept your manuscript and schedule it for publication until you have made the required changes.

PRESS

Sincerely, 

Gabriel Gasque, Ph.D. 

Senior Editor 

PLOS Biology

ggasque@plos.org